# A conserved regulator controls asexual sporulation in the fungal pathogen *Candida albicans*

Arturo Hernández-Cervantes[1,13], Sadri Znaidi [1,2,3], Lasse van Wijlick [1], Iryna Denega [1,4], Virginia Basso [1,4,14], Jeanne Ropars [1,15], Natacha Sertour[1], Derek Sullivan [5], Gary Moran[5], Louise Basmaciyan[6,7], Fabienne Bon[6], Frédéric Dalle[6,7], Marie-Elisabeth Bougnoux[1,8], Teun Boekhout [9,10], Ying Yang[11], Zongwei Li[12], Sophie Bachellier-Bassi [1✉] & Christophe d'Enfert [1✉]

Transcription factor Rme1 is conserved among ascomycetes and regulates meiosis and pseudohyphal growth in *Saccharomyces cerevisiae*. The genome of the meiosis-defective pathogen *Candida albicans* encodes an Rme1 homolog that is part of a transcriptional circuitry controlling hyphal growth. Here, we use chromatin immunoprecipitation and genome-wide expression analyses to study a possible role of Rme1 in *C. albicans* morphogenesis. We find that Rme1 binds upstream and activates the expression of genes that are upregulated during chlamydosporulation, an asexual process leading to formation of large, spherical, thick-walled cells during nutrient starvation. *RME1* deletion abolishes chlamydosporulation in three *Candida* species, whereas its overexpression bypasses the requirement for chlamydosporulation cues and regulators. *RME1* expression levels correlate with chlamydosporulation efficiency across clinical isolates. Interestingly, *RME1* displays a biphasic pattern of expression, with a first phase independent of Rme1 function and dependent on chlamydospore-inducing cues, and a second phase dependent on Rme1 function and independent of chlamydospore-inducing cues. Our results indicate that Rme1 plays a central role in chlamydospore development in *Candida* species.

[1] Unité Biologie et Pathogénicité Fongiques, Institut Pasteur, USC, 2019 INRA Paris, France. [2] Institut Pasteur de Tunis, Laboratoire de Microbiologie Moléculaire, Vaccinologie et Développement Biotechnologique, Bâtiment Etienne Burnet, 13 Place Pasteur, B.P.74, 1002 Tunis-Belvédère, Tunisia. [3] University of Tunis-El Manar, 1036 Tunis, Tunisia. [4] Université de Paris, Sorbonne Paris Cité, Paris, France. [5] Dublin Dental University Hospital and School of Dental Science, Trinity College Dublin, Dublin 2, Ireland. [6] UMR PAM, Université de Bourgogne Franche-Comté, AgroSup Dijon – Equipe VAIMiS, Dijon, France. [7] Centre Hospitalier Universitaire François Mitterand, Service de Parasitologie Mycologie, Dijon, France. [8] Unité de Parasitologie-Mycologie, Service de Microbiologie clinique, Hôpital Necker-Enfants-Malades, Assistance Publique des Hôpitaux de Paris (APHP), Université de Paris, Paris, France. [9] Westerdijk Fungal Biodiversity Institute, Utrecht, The Netherlands. [10] Institute of Biodiversity and Ecosystem Dynamics (IBED), University of Amsterdam, Amsterdam, The Netherlands. [11] Beijing Institute of Radiation Medicine, Beijing 100850, China. [12] Center for Hospital Infection Control, Institute for Disease Control & Prevention, Beijing 100071, China. [13] Present address: Université de Paris / Inserm, 1 Avenue Claude Vellefaux, F-75010 Paris, France. [14] Present address: Department of Pathology and Laboratory Medicine, Norris Comprehensive Cancer Center, Keck School of Medicine, University of Southern California, Los Angeles, CA, USA. [15] Present address: Université Paris-Saclay, CNRS, AgroParisTech, Écologie, Systématique, Évolution, F-91405 Orsay, France. ✉email: sophie.bachellier-bassi@pasteur.fr; christophe.denfert@pasteur.fr

Some microorganisms have the ability to survive nutrient limitations in their environments through a complex biological process called "sporulation". Fungi form spores as a way to subsist under non-favourable conditions, but also as part of asexual and sexual reproductive cycles. In general, fungi reproduce sexually to generate genetic variability in the population, therefore enhancing survival in new environments. For instance, under harsh growth conditions, the budding yeast *Saccharomyces cerevisiae* undergoes meiosis, leading to the formation of four haploid spores that upon the improvement of the environmental conditions will develop into Mat**a** and Matα haploid cells, able to mate and go through the sexual cycle[1]. However, asexual sporulation is the main reproductive mode of many fungi, which form different types of spores that vary in function, shape, colour, size, and number. Chlamydospores are enlarged thick-walled cells produced within hyphae or at hyphal tips, found in several ascomycetous and basidiomycetous fungi, as, for example, *Dibotryon morbosum*, a phytopathogen causing the black knot disease[2], the nematode-trapping fungus *Duddingtonia flagrans*[3,4], and the human pathogenic fungus *Cryptococcus neoformans*[5]. Chlamydospores exhibit very diverse and sometimes elusive functions. In *Fusarium* species, chlamydospores allow for long periods of survival in soil under adverse conditions, but they are also used to infect their host plants[6]; in *Paracoccidioides brasiliensis* they are an intermediate morphotype necessary for the hypha-yeast transition[7]; in *Cryptococcus neoformans*, chlamydospores are linked to monokaryotic fruiting and mating[5]. Among the species of the genus *Candida*, only *C. albicans* and *C. dubliniensis*, two closely related species[8], produce chlamydospores[9] which can germinate into yeast cells, but do not resist environmental stresses during long periods[10]. There is little evidence regarding the role of chlamydospores in the pathogenicity and life cycle of *C. albicans* and *C. dubliniensis*. They are produced in vitro under harsh growth conditions, such as poor medium, low temperature, microaerophilia, and a few studies have reported the presence of chlamydospore-like cells in samples isolated from patients with candidiasis[11–15] or from tissues of mice infected with *C. albicans*[16,17], therefore suggesting a potential role of chlamydospores during infection. Besides, a handful of studies have shown that chlamydosporulation in *C. albicans* is controlled by the transcriptional regulators of filamentation Efg1 and Nrg1, the stress-activated protein kinase Hog1, as well as by the TOR and Ras-cAMP-PKA signalling cascades[18–21]. Although these studies have shed light on their formation process, it remains unclear how these signalling pathways operate to form chlamydospores.

In this study, we discover that chlamydospore-forming *Candida* species, including *C. albicans*, *C. dubliniensis* and *C. buenavistaensis*, rely on Rme1 to develop into chlamydospores. We show that *C. albicans* Rme1 directly modulates the expression of chlamydospore-specific genes and that its deletion or overexpression results in the absence or abundant formation of chlamydospores, respectively. Consistently, *RME1* expression levels correlate with chlamydosporulation efficiency in *C. albicans* clinical isolates. We also demonstrate that Rme1 function in chlamydospore formation bypasses the requirement for other known regulators of this process, including Efg1 and Hog1. Building on our previous findings that *RME1* is part of the Sfl1/Sfl2 transcriptional circuitry that controls *C. albicans* morphogenesis, we reveal that two regulators of *RME1* expression, Sfl1 and Ndt80, are also involved in chlamydospore formation. Because Rme1, a repressor of meiosis in *S. cerevisiae*, controls a developmental program leading to asexual spore formation in the meiosis-defective species *C. albicans*, our work pinpoints a striking example of transcriptional rewiring and reflects the extended diversity of the mechanisms behind evolutionary tinkering among species of the ascomycete lineage.

## Results

**Rme1 regulates chlamydosporulation in *C. albicans*.** Previous work in our laboratory identified the *RME1* gene as a direct target of Sfl1, a repressor of *C. albicans* hyphal development[22]. In *S. cerevisiae*, Rme1 is involved in regulating meiosis[23], a process that has been lost in *C. albicans*. Thus, we sought to explore the function of *C. albicans* Rme1 and generated *C. albicans* strains that over-produced either an intact- or an N-terminally rTAP-tagged Rme1 protein in a doxycycline-dependent manner (all strains are listed in Supplementary Data 1). The expression of rTAP-Rme1 was confirmed by Western blotting (Supplementary Fig. 1a). We also observed the nuclear localization of a $P_{TET}$–controlled N-terminal fusion of the GFP to Rme1 in $P_{TET}$-inducing conditions (Supplementary Fig. 1b). Overexpression of either *RME1* or *rTAP-RME1* led to pseudohyphal formation in rich liquid medium at 30 °C (Supplementary Fig. 1c), suggesting a role of Rme1 in the regulation of morphogenesis and indicating that N-terminal epitope-tagging did not affect Rme1 functionality.

In order to identify Rme1 binding sites on the *C. albicans* genome, we conducted ChIP-chip experiments using the *RME1* (control)- and *rTAP-RME1*(tagged)-overexpressing strains after 4 h of induction with doxycycline. Using the CisGenome peak algorithm[24], we identified 724 5′-regulatory regions occupied by Rme1 (Supplementary Data 3–5). Unexpectedly, Rme1 was predominantly bound to the 5′-regulatory regions of genes upregulated during chlamydospore formation (e.g. *CSP1*, *CSP2*, *ORF19.654*, *ORF19.555*, *ORF19.4463*, *PGA55*)[25], as well as at the promoter regions of genes whose products are involved in cell wall biosynthesis, oxidative stress and filamentation. Rme1 was also bound to its own promoter, suggesting a transcriptional autoregulation (Fig. 1a). We validated the ChIP-chip data by ChIP-qPCR, detecting Rme1 binding at the upstream regions of a set of selected genes (Supplementary Fig. 2a).

In parallel, we performed global gene expression profiling of the *RME1* overexpression strain grown for 2 and 4 h in doxycycline-containing medium as compared to the non-induced culture (Supplementary Data 6). We found 423 and 446 upregulated (fold-change ≥1.5, $P < 0.05$) and 185 and 235 downregulated (fold-change ≤ −1.5, $P < 0.05$) genes after 2 and 4 h of induction, respectively (Fig. 1b, c). As a control, doxycycline treatment alone did not significantly alter gene expression[26] (GSE67226, https://www.ncbi.nlm.nih.gov/geo/query/acc.cgi?acc=GSE67226). In agreement with the ChIP-chip data, we observed that the most upregulated genes at both time points were those encoding chlamydospore markers (e.g. *CSP1*, *ORF19.654*, *ORF19.555*, *ORF19.4463*, *PGA55*, *CSP2*, Supplementary Data 6). *RME1* overexpression also triggered the upregulation of genes involved in cell wall biosynthesis, filamentation, glyoxysomal metabolism, mating, and white-opaque switching (Fig. 1c). Genes involved in DNA replication/repair/recombination and cell-cycle progression, and in DNA methylation/demethylation, mRNA transport, transcription, filamentation and cell wall biosynthesis were downregulated upon *RME1* overexpression. The transcriptomics results were confirmed by RT-qPCR, measuring the gene expression levels of a subset of up- and downregulated genes (Supplementary Fig. 2b).

The overlap between genes upregulated at 2 and 4 h and bound by Rme1 corresponded predominantly to genes encoding chlamydospore markers, or involved in cell wall biosynthesis, oxidative stress and filamentation (Fig. 1c), suggesting that Rme1 directly regulates these processes. We further verified the association between Rme1 transcriptional targets and the chlamydospore developmental program by looking at the occurrence of genes that are both bound and upregulated by Rme1 (257 out of 6.083 *C. albicans* genes represented on the

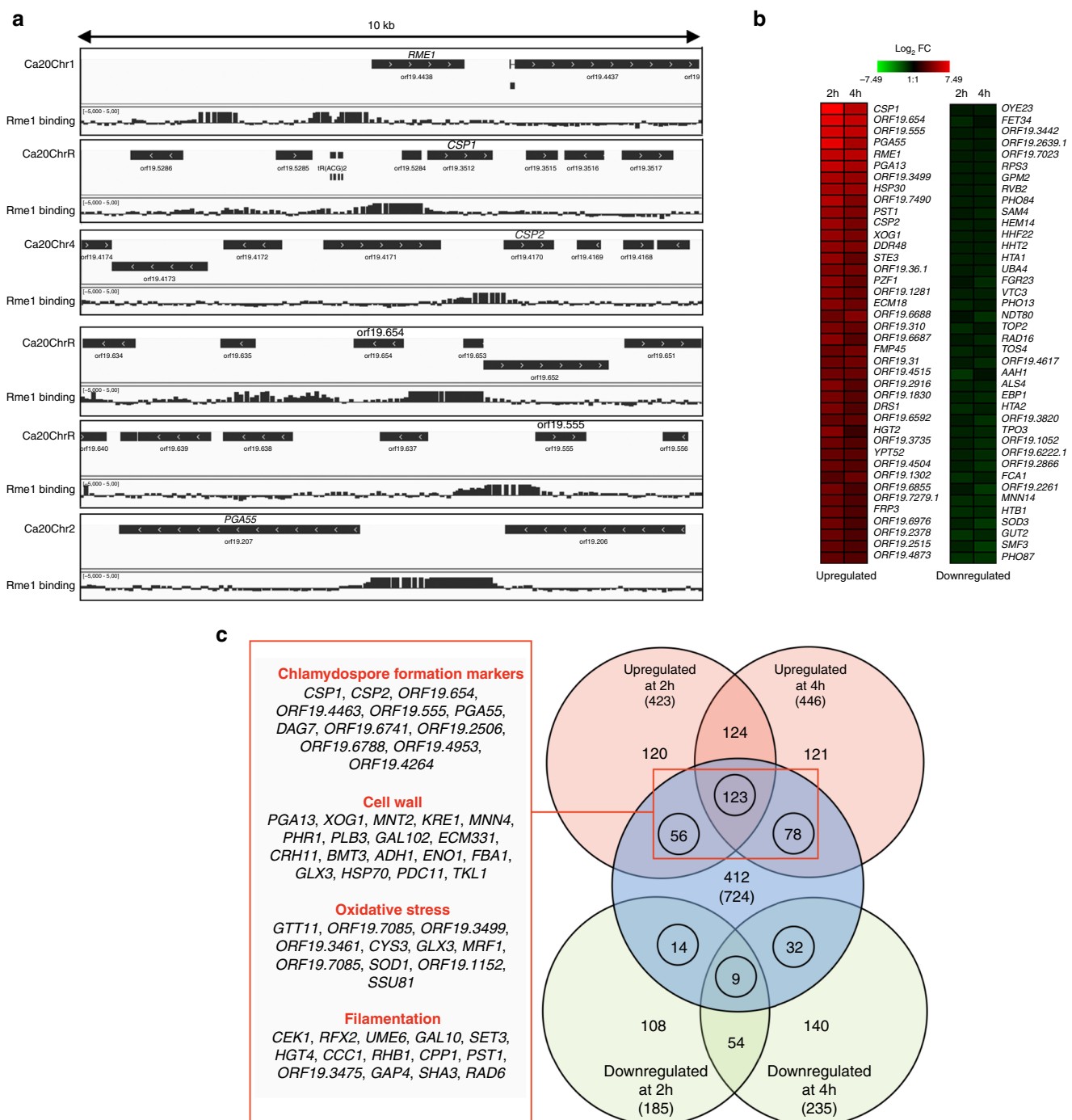

**Fig. 1 *C. albicans* Rme1 is a positive regulator of chlamydospore formation markers. a** Rme1 occupancies along 10-kb intervals of selected locations from the *C. albicans* genome (Assembly 20, the corresponding chromosome numbers are indicated on the left of each panel). The relative signal intensities of the 60-bp probes covering the whole *C. albicans* genome following enrichment of the rTAP-tagged Rme1-coimmunoprecipitated DNA relative to DNA from a mock immunoprecipitation (in an untagged strain background) are plotted. Data from one ChIP-chip experiment out of two are shown. The orientation of each ORF is depicted by the arrows in the black rectangles. Binding maps were generated using the IGV genome browser[61]. **b** Heat map showing the expression pattern in $P_{TET}$-*RME1* transcript profiling data at time points 2 and 4 h after induction with 40 μg/mL doxycycline. The heat map was generated with Genesis version 1.7.6[74] using the average expression level of three replicates. Genes were ranked according to average expression value between the two time points. **c** Venn diagram showing the overlap between genes transcriptionally modulated by $P_{TET}$-*RME1* at time points 2 and 4 h (gene expression fold-change ≥1.5 or ≤−1.5; $P < 0.05$, two-tailed Welch's *t*-test, $n = 3$ independent biological samples) and bound by Rme1. Numbers in Venn diagram indicate the number of genes and those between parentheses indicate the total number of upregulated (light red circles), downregulated (light green circles) and bound (light blue circle) genes. Numbers within the circles indicate the number of genes that are both bound and transcriptionally modulated by Rme1. The name of few genes and their functional categories are shown in the linked red box (55 out of 257 bound and upregulated genes are shown). The ChIP-chip experiments was performed with two independent biological replicates, over two experiments.

arrays, 4.2%, Fig. 1c) among the set of genes that were found by Palige et al.[25] to be upregulated ≥ 2 fold in the C. albicans nrg1Δ/nrg1Δ mutant grown in the chlamydospore-inducing Staib medium. We found 34 occurrences out of 271 genes upregulated in the nrg1Δ/nrg1Δ strain (12.5%), yielding a ~3 fold enrichment of Rme1 direct targets in Palige et al. datasets ($P = 6.72 \times 10^{-9}$ using a hypergeometric test).

**Rme1 controls chlamydosporulation in species other than C. albicans.** The data presented above raised the question of whether Rme1 acts as a regulator of chlamydosporulation. We tested this hypothesis by growing the C. albicans wild-type (WT) strain SC5314, an rme1 knockout mutant (rme1ΔΔ, CEC4411) and an RME1 overexpression strain ($P_{TET}$-RME1 in a WT background, CEC5031) in "chlamydospore-inducing conditions" (on Potato Carrot Bile (PCB) solid medium at 25 °C in darkness and microaerophilia)—with doxycycline for $P_{TET}$ induction—for 3 days, a condition that triggers the formation of chlamydospores by the SC5314 WT strain (Fig. 2a). Strikingly the $P_{TET}$-RME1 strain formed numerous rounded cells with a chlamydospore-like aspect in the presence of doxycycline, whereas the rme1ΔΔ strain did not produce chlamydospores (Fig. 2a), indicating a role of RME1 in the control of chlamydosporulation. The rme1ΔΔ mutant was also tested in a systemic infection mouse model, and for mating, but did not exhibit any defect (data not shown). Interestingly, when grown on PCB medium for 3 days at 37 °C in the dark and under microaerophilia, the WT, rme1ΔΔ and $P_{TET}$-RME1 strains did not form chlamydospores but rather underwent filamentous growth (data not shown).

Palige et al.[25] identified chlamydospore-specific markers in C. albicans, including CSP1, encoding a chlamydospore-specific cell wall protein, and PGA55. In order to confirm that the rounded cells formed upon RME1 overexpression were genuine chlamydospores, we introduced a CSP1-GFP fusion or a $P_{PGA55}$-GFP fusion in the WT, rme1ΔΔ and $P_{TET}$-RME1 strains. RME1 overexpression resulted in the massive production of PGA55-expressing or Csp1-GFP surface-decorated rounded cells, consistent with these cells being chlamydospores (Fig. 2b). No such cells could be detected when RME1 was lacking (Fig. 2b). We then reasoned that the high lipid content of chlamydospores[27] would allow staining by BODIPY, a lipophilic fluorescent dye[28]. Similarly, EosinY, which detects chitosan, a component of S. cerevisiae's ascospores- and C. neoformans chlamydospores-cell wall[29], and does not stain viable C. albicans blastospores[30], could detect specific features of the C. albicans chlamydospores cell wall. Both EosinY and BODIPY stained the rounded cells produced by the WT and $P_{TET}$-RME1 strains, further highlighting features associated with chlamydospore formation (Fig. 2c), thus allowing confirmation of chlamydospore formation in clinical isolates.

C. dubliniensis is the only yeast species other than C. albicans known to form chlamydospores when grown on some nutrient-poor media[9]. Considering that these two species are closely related and share chlamydospore-specific gene expression profiles[25], we constructed an rme1ΔΔ mutant in C. dubliniensis and observed that chlamydosporulation was abolished in this mutant (Fig. 2d, left panels). Candida buenavistaensis, a species isolated from an insect, is phylogenetically closely related to C. albicans and C. dubliniensis[31], so we sought to determine if RME1 would exhibit the same function in this species. To this aim, we constructed a C. buenavistaensis rme1ΔΔ mutant and compared it to the WT strain after growth in liquid chlamydospore-inducing conditions. The WT exhibited round-shaped structures reminiscent of chlamydospores, which were absent from the rme1ΔΔ strain (Fig. 2d). Thus, the function of Rme1 in controlling chlamydosporulation is

shared between C. albicans and C. dubliniensis, and also in the closely related species C. buenavistaensis.

**Chlamydosporulation depends on Rme1 species-specificity.** Since Rme1 is a central regulator of chlamydospore formation in chlamydospore-forming Candida species, we speculated whether the differentiation process would rely mostly on Rme1, or if other factors would be needed. To this aim we overexpressed RME1 from either chlamydospore-forming (C. dubliniensis, Cd; and C. buenavistaensis, Cb) or -non-forming (C. tropicalis, Ct; and C. parapsilosis, Cp) strains in the C. albicans rme1ΔΔ mutant. The overexpression of CdRME1 and CbRME1 restored C. albicans ability to form numerous chlamydospores on doxycycline-containing PCB (Fig. 2e), whereas overexpressing RME1 from C. tropicalis or C. parapsilosis induced pseudohyphal growth (Fig. 2e). This unambiguously shows that the ability to trigger chlamydosporulation is linked to the Rme1 protein. In rich medium, however, the overexpression of all heterologous RME1 except that of C. buenavistaensis induced pseudohyphal formation (not shown).

**RME1 and its targets expression levels in C. albicans natural isolates.** As mentioned above and evidenced in Fig. 2a, the reference strain SC5314 standardly used for gene-function studies in C. albicans is a rather poor chlamydospore former. Other studies have stressed that the efficiency of chlamydospore production may vary between C. albicans isolates[32–34], and we hypothesized that this may depend upon Rme1 activity.

In order to test this hypothesis, we first scored a collection of 149 commensal and clinical C. albicans isolates for their ability to form chlamydospores in liquid chlamydospore-inducing conditions (Supplementary Data 7 and Supplementary Fig. 4). Under the tested conditions, 7 isolates formed chlamydospores profusely (score = 4–5), 10 isolates showed an intermediate phenotype (score = 2–3) and 132 isolates formed chlamydospores scarcely, or not at all (score = 0–1). It is worth mentioning that, as SC5314, most of these strains were able to form chlamydospores when tested on solid chlamydospore-inducing conditions (not shown).

The expression levels of RME1 and some Rme1 targets, namely CSP1, CSP2, PGA55 and ORF19.654, were measured by RT-qPCR, in isolates with profuse (CEC3620, CEC3683, CEC2018 and CEC4039) or absent (CEC1424, CEC1426 and CEC3539) chlamydospore formation, taking as a reference the WT SC5314 strain (Fig. 3a). We observed that the expression levels of RME1 and its targets were significantly higher in the isolates with high scores than in the WT strain and the isolates with low scores (Fig. 3b). Hence, the strong correlation between the expression levels of RME1 and its targets and the ability of C. albicans to form chlamydospores is consistent with a central function of Rme1 in regulating C. albicans chlamydosporulation.

**RME1 is necessary for the expression of chlamydospore-related genes.** A possible limitation of the overexpression approach that we used to define the Rme1 regulon lies in condition-independent and/or potential off-target effects. Thus, we sought to complement this approach by assessing the impact of RME1 inactivation on the transcriptome when C. albicans was grown under conditions conducive to chlamydosporulation. To this aim, we used an rme1ΔΔ derivative of C. albicans strain CEC2018, a clinical isolate efficiently producing chlamydospores in liquid PCB medium (Fig. 3a, Supplementary Data 7). The CEC2018-rme1ΔΔ knockout mutant was defective for chlamydospore formation, PGA55 expression and Csp1 production (Fig. 4a). Comparison of the transcript profiles of strains CEC2018 and CEC2018-rme1ΔΔ grown for 24 h in liquid PCB medium identified 122 downregulated- (fold-change ≤ −1.5, P <

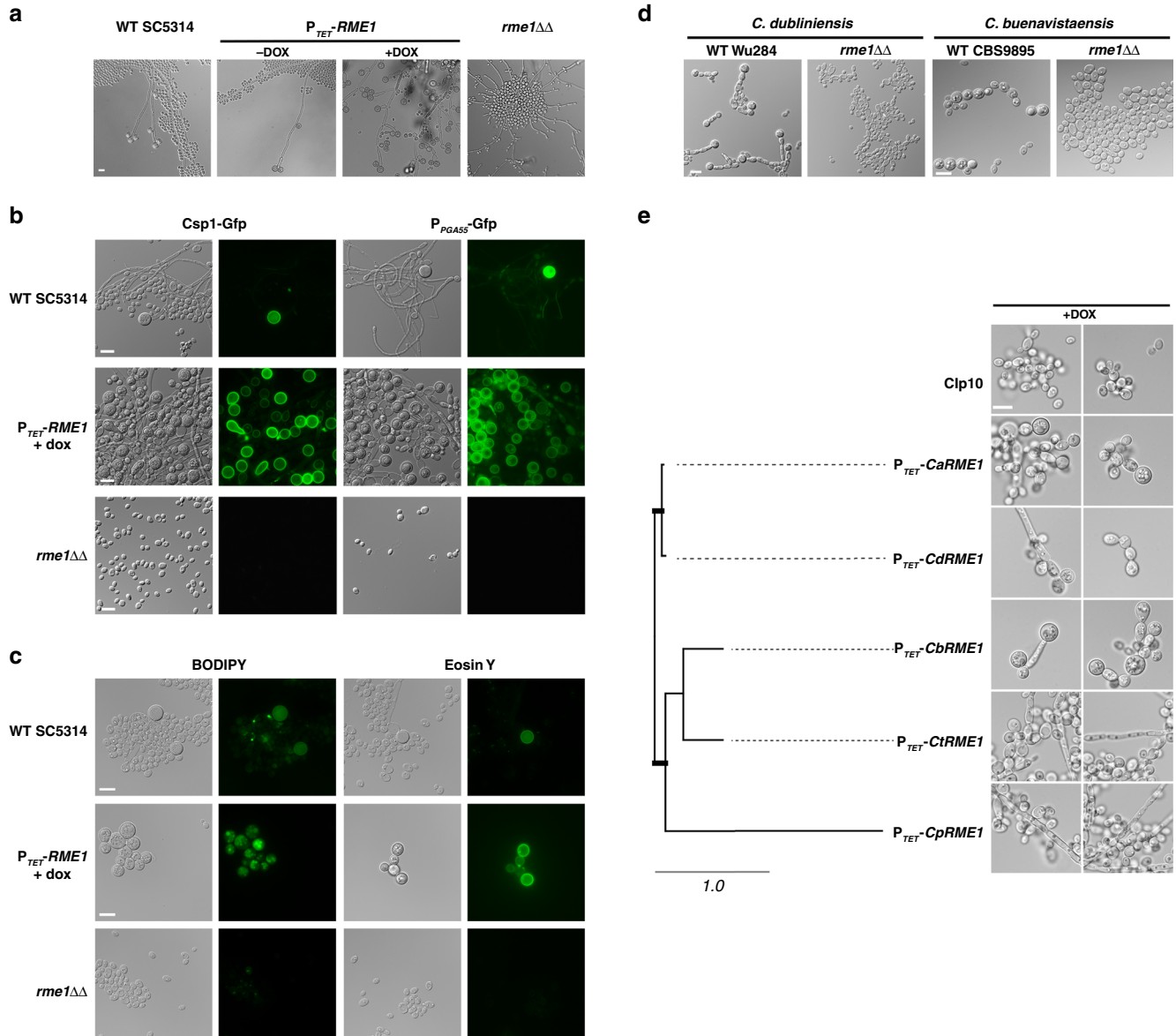

**Fig. 2 RME1 controls chlamydospore formation. a** *C. albicans* WT SC5314, P$_{TET}$-*RME1* (CEC5031) and *rme1ΔΔ* (CEC4694) strains growing on PCB agar plates. Photomicrographs were taken after 3 days of incubation at 25 °C in the dark and under microaerophilic conditions. Scale bar = 10 μm, applies to all pictures. **b** Both Csp1 and Pga55 expressions are dependent on Rme1. *C. albicans* WT SC5314, P$_{TET}$-*RME1* and *rme1ΔΔ* strains expressing either the *CSP1-GFP* or P$_{PGA55}$-*GFP* fusions (CEC4783, CEC4827, CEC4788, CEC4791, CEC4967 and CEC4804, respectively) were cultured on a cellophane film covering PCB plates at 25 °C, in the dark and under microaerophilic conditions for 3 days. Cells were recovered from the cellophane film and inspected by phase contrast and fluorescence microscopy. Induction of Rme1 was achieved by adding doxycycline to the medium to a final concentration of 40 μg/mL. Scale bars = 10 μm. **c** *C. albicans* WT SC5314, P$_{TET}$-*RME1* (CEC4741) and *rme1ΔΔ* (CEC4694) strains grown for 2–6 days in PCB medium supplemented with doxycycline for inducing *RME1* overexpression were incubated with BODIPY or Eosin Y. Scale bars = 10 μm. **d** *C. dubliniensis* WT Wü284 and its *rme1ΔΔ* derivative (left panels) and *C. buenavistaensis* CBS9895 and its *rme1ΔΔ* derivative (right panels) were grown in PCB liquid medium at 25 °C, in the dark, during 24 h. Scale bars = 10 μm. **e** *C. albicans rme1ΔΔ* strains with conditional heterologous *RME1* overexpression were grown overnight in liquid doxycycline-containing PCB, in the dark, at 25 °C. CaRME1: *C. albicans*, CdRME1: *C. dubliniensis*, CbRME1: *C. buenavistaensis* (all chlamydospore proficient species), CtRME1: *C. tropicalis* and CpRME1: *C. parapsilosis* (chlamydospore deficient species). Two representative fields are presented for each strain, all images are at the same scale. Scale bar = 10 μm. The tree has been generated with PHYML[72] from a multiple alignment of Rme1 protein sequences done with MUSCLE[69] (Supplementary Fig. 3). Thick bars represent bootstrap supports >98% (bootstrap analysis of 100 resampled datasets); branch lengths are shown and the scale bar represents 1 substitution per site.

0.05) and 26 upregulated-genes (fold-change ≥1.5, *P* < 0.05) when *RME1* was knocked out (Supplementary Data 8). Consistent with our previous results, we observed that, under chlamydospore-inducing conditions, Rme1 controls the expression of chlamydospore formation markers, namely *CSP1* and *CSP2*, as well as three closely related ORFs, *ORF19.654*, *ORF19.4463* and *ORF19.555* (Fig. 4b). Interestingly, many of the

downregulated genes were linked to filamentation (*CEK1*, *ORF19.6817*, *CCC1*, *CUP9*, *ALD5*, *ORF19.4456*, *SOD1* and *PDI1*), supporting the notion that Rme1 controls filamentation when *C. albicans* is growing under chlamydospore-inducing conditions. These transcriptomics data were confirmed by RT-qPCR, measuring gene expression levels of a subset of up- and down-regulated genes (Supplementary Fig. 5).

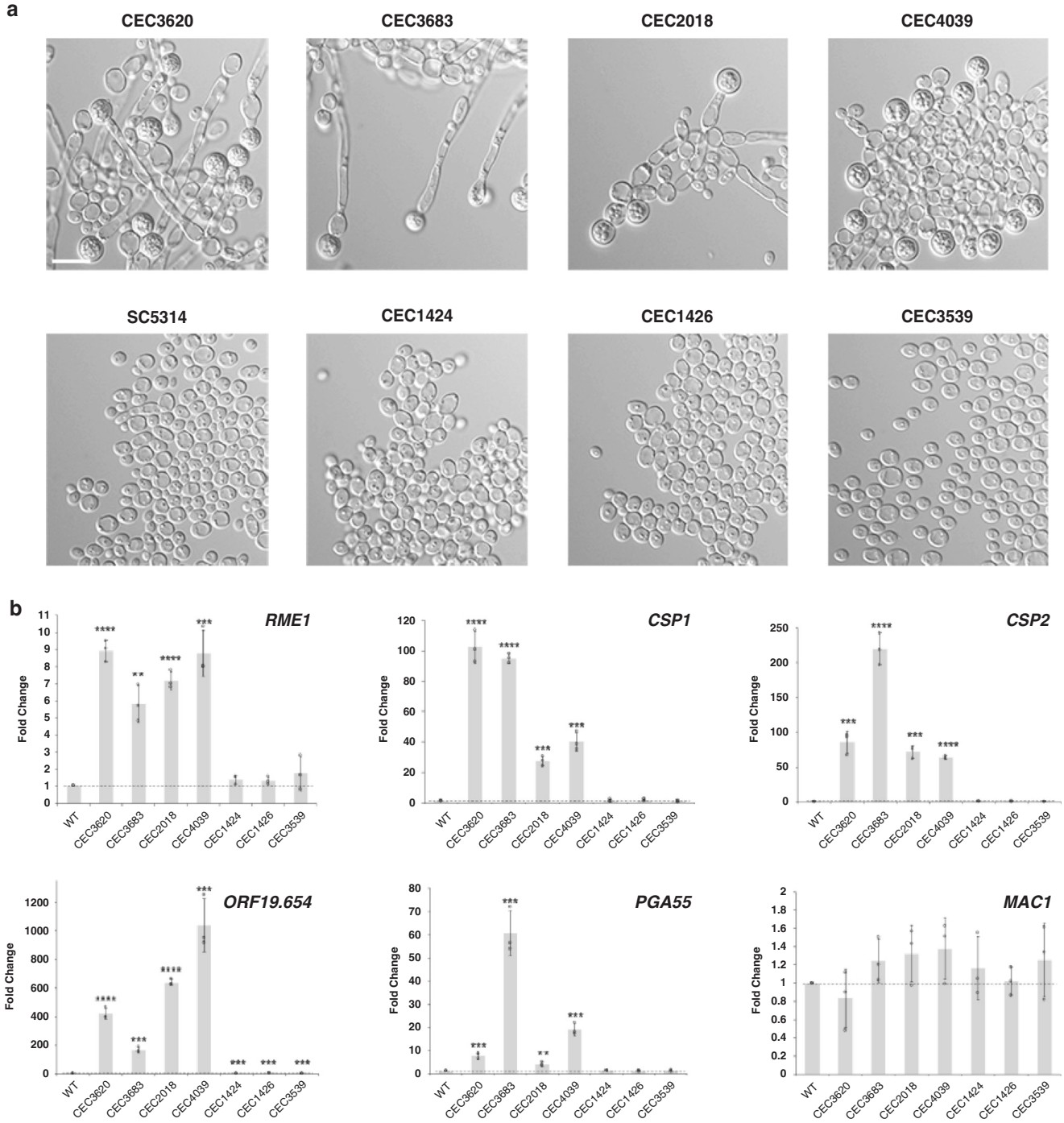

**Fig. 3 Expression levels of *RME1* and its targets are related to the capacity to form chlamydospores in *C. albicans* strains. a** Clinical isolates of *C. albicans* exhibit differences in chlamydospore formation efficiency. The upper panel shows strains that formed masses of chlamydospores when grown in liquid PCB at 25 °C for 24 h, whilst the bottom panel shows strains defective for chlamydospore formation under the tested conditions. WT SC5314 strain was used as reference for comparison. Scale bar = 10 μm. **b** Expression levels of *RME1* ($P = 0.000022$; 0.001395; 0.000035; 0.00062; 0.058885; 0.083474; 0.275589), *CSP1* ($P = 0.000074$; <0.000001; 0.000126; 0.000357; 0.71581; 0.156621; 0.221736), *CSP2* ($P = 0.000797$; 0.000078; 0.000163; 0.000001; 0.848682; 0.388113; 0.130289), *ORF19.654* ($P = 0.000047$; 0.000104; <0.000001; 0.00062; 0.000517; 0.000496; 0.000658), *PGA55* ($P = 0.000905$; 0.000443; 0.005717; 0.000285; 0.092887; 0.52506; 0.330046) and *MAC1* (control) were quantified by RT-qPCR in *C. albicans* strains and normalized against SC5314 values. Data are expressed as the mean ± SD, $n = 3$ biological replicates, over three experiments. *$P < 0.05$; **$P < 0.01$; ***$P < 0.001$; ****$P < 0.0001$.

We compared the list of genes that are positively regulated by Rme1 in P$_{TET}$-*RME1* transcriptomics data to those downregulated in CEC2018-*rme1*ΔΔ data. We found an overlap of 58 genes, many of which are involved in chlamydospore formation and filamentation (Fig. 4b, c). Interestingly, many genes were upregulated only upon *RME1* overexpression (either involved in carbohydrate metabolic processes or encoding cell wall-associated proteins), suggesting that Rme1 could regulate other cellular processes besides chlamydosporulation when *C. albicans* is grown under conditions not conducive to chlamydospore formation.

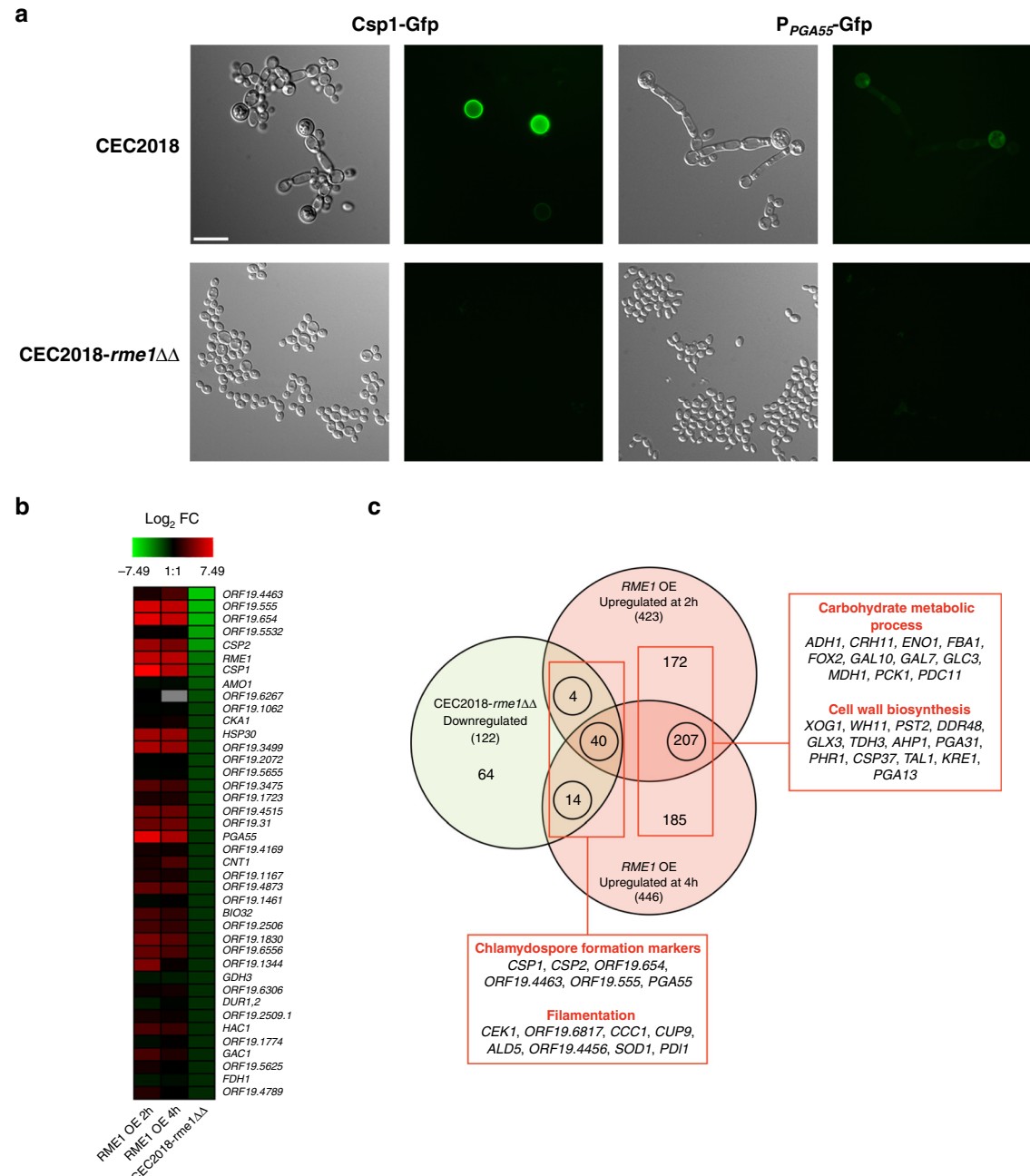

**Fig. 4 _RME1_ disruption in the clinical isolate CEC2018 confirms Rme1 involvement in chlamydosporulation. a** Strain CEC2018 and the CEC2018-_rme1ΔΔ_ derivative expressing the _CSP1-GFP_ (left panels) and P_PGA55_-_GFP_ (right panels) fusions were cultured in PCB liquid medium at 25 °C for 24 h and then inspected by phase contrast and fluorescence microscopy. Scale bar = 10 μm. **b** Heat map comparing the downregulated genes in the transcriptomic analysis performed in the CEC2018-_rme1ΔΔ_ strain (gene expression fold-change ≤ −1.5; _P_ < 0.05, _n_ = 3 independent biological replicates) and the upregulated genes at 2 h and 4 h in the expression profiling in the strain expressing P_TET_-_RME1_ (gene expression fold-change ≥1.5; _P_ < 0.05; statistical significance was assessed using a two-tailed Welch's _t_-test, _n_ = 3 independent biological samples). **c** Venn diagram showing the overlap between the upregulated genes by P_TET_-_RME1_ at 2 h and 4 h (gene expression fold-change ≥1.5; _P_ < 0.05, _n_ = 3 independent biological replicates) and the downregulated genes (gene expression fold-change ≤ −1.5; _P_ < 0.05, _n_ = 3 independent biological replicates) in the CEC2018-_rme1ΔΔ_ strain. Numbers in Venn diagram represent the number of modulated genes and those between parentheses indicate the total number of upregulated genes upon _RME1_ overexpression (light red circles) and downregulated ones in the _rme1ΔΔ_ mutant (light green circle). Circled numbers represent the overlap of transcriptionally modulated genes in both experiments. The linked red box shows the name of a few genes and their corresponding functional categories. Statistical significance was assessed using a two-tailed Welch's _t_-test.

**Biphasic expression of _RME1_ and its autoregulation.** As previously described, our ChIP-chip data suggested a transcriptional autoregulation for Rme1. To test this hypothesis, an assay was conducted to evaluate if expressing Rme1 ectopically in either the WT SC5314 or _rme1ΔΔ_ background strains could promote a regulatory loop during growth in rich medium (YPD, see "Methods"). We observed that a 3 h-pulse of doxycycline in YPD was sufficient to turn on the chlamydospore formation pathway in both backgrounds (Fig. 5a). When both strains were transferred to fresh YPD without doxycycline, only the WT strain

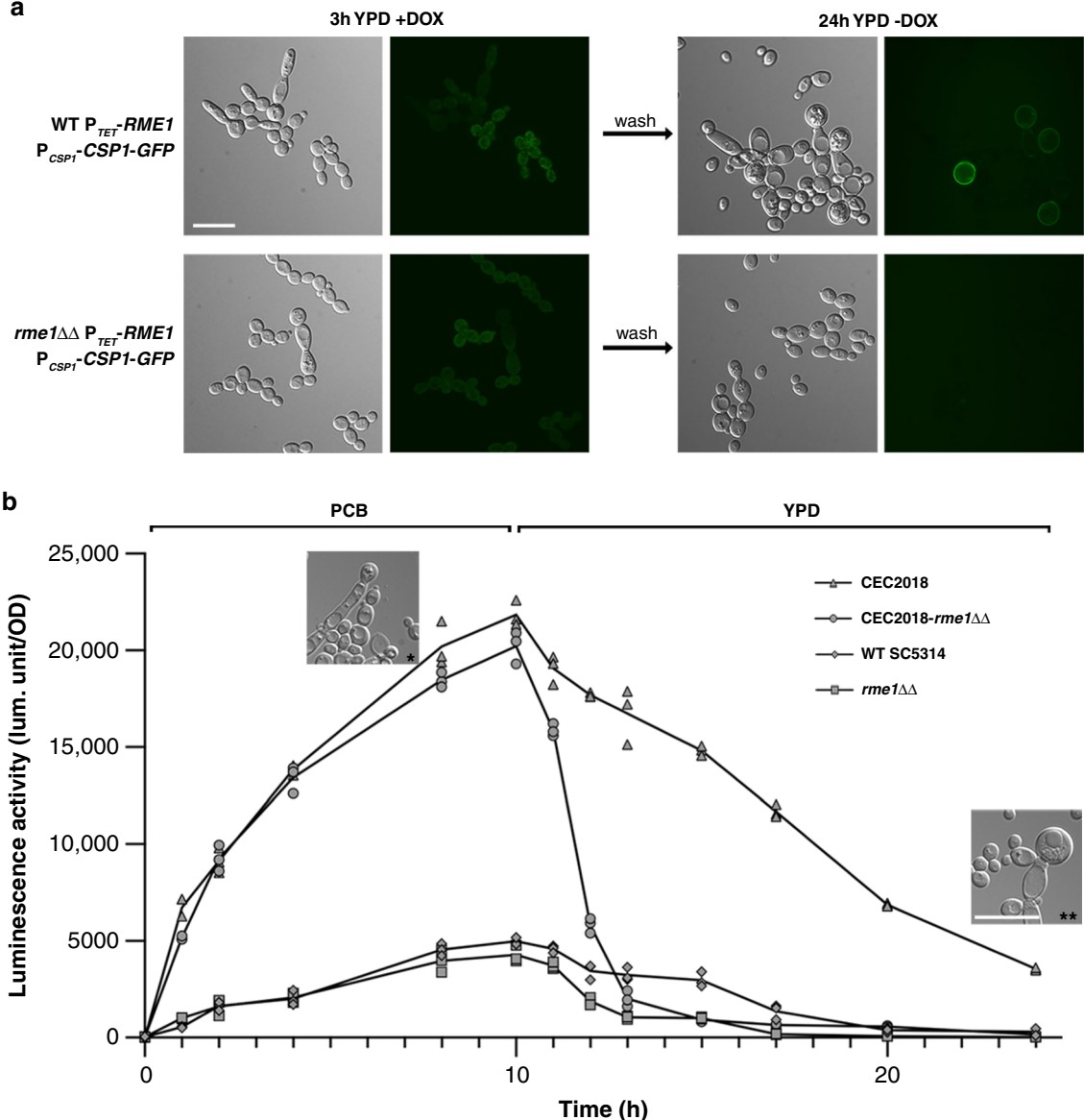

**Fig. 5 Rme1 positively regulates its own expression. a** *RME1* native promoter is positively regulated by the ectopic expression of *RME1* in *C. albicans* WT strain. *C. albicans* WT and *rme1ΔΔ* strains expressing P$_{TET}$-*RME1* and *CSP1-GFP* (CEC4827 and CEC4789, respectively) were grown in liquid doxycycline-containing YPD (+dox) at 30 °C for 3 h, then cells were centrifuged, washed and resuspended in fresh YPD without doxycycline and incubated overnight at 30 °C. Scale bar = 10 μm. **b** *RME1* expression is controlled by Rme1 endogenously expressed under harsh conditions. Luciferase activity levels were measured in *C. albicans* strains expressing the P$_{RME1}$-*gLUC* fusion. After a 10-h incubation at 25 °C in liquid PCB medium, cells were transferred to fresh YPD medium and incubated for 14 h. Aliquots were taken at the indicated time points (*x*-axis) and the data presented are the recorded luciferase activity levels (*y*-axis). The values of three independent replicates are plotted (*n* = 3). Inset pictures show CEC2018 cells after a 10-h incubation in PCB (*) or after the following overnight incubation in YPD (**). Both images are at the same scale. Scale bar = 10 μm.

could sustain chlamydospore formation through time, forming mature chlamydospores detected by the Csp1-Gfp fusion after an overnight incubation at 30 °C, whereas the *rme1ΔΔ* strain only grew as yeasts (Fig. 5a). Thus, the pulse of Rme1 produced during doxycycline induction was able to initiate the autoregulatory loop in the WT strain, resulting in efficient chlamydospore production even in rich medium and at a non-permissive temperature.

In order to get further insight into *RME1* regulation, we created a P$_{RME1}$-*gLUC* fusion allowing quantification of *RME1* promoter expression levels by means of luciferase activity upon shifting from rich- to poor-nutrient growth conditions. Because the WT SC5314 strain scarcely forms chlamydospores in liquid PCB medium, we chose to quantify luciferase activity in the CEC2018 background (CEC4870) as well as in the *rme1ΔΔ* derivative

(CEC4851). The CEC2018 strain forms protochlamydospores after 10 h of incubation in PCB (see inset picture in Fig. 5b), therefore, this time point was selected to transfer the cells to YPD medium and culture them for up to 24 h. The luciferase activity levels measured in the CEC2018-*rme1ΔΔ* strain dropped drastically when cells were transferred to rich medium, while the luciferase activity was observed much longer in the WT CEC4870, which formed mature chlamydospores after the overnight incubation in YPD (see inset picture in Fig. 5b).

These results indicate that Rme1 endogenously expressed during 10 h of incubation in PCB is able to sustain the *RME1* promoter induction during growth in rich medium, and therefore form mature chlamydospores in unfavourable conditions. The WT SC5314 strain showed lower luciferase activity levels as

compared to those exhibited by the CEC2018 strain, thereby making a comparison with the derivative rme1ΔΔ mutant much more difficult. Taken together, the results from the two different approaches confirm that Rme1 positively regulates its own expression.

**RME1 overexpression bypasses the requirement for other regulators**. Among the genes influencing chlamydospore formation, *EFG1* and *HOG1* have been shown to be essential to this developmental pathway[18,35,36]. *HOG1* encodes a mitogen-activated protein (MAP) kinase that mediates an adaptive response to osmotic and oxidative stress, cell wall biosynthesis and chlamydospore formation[18,36,37], while *EFG1* encodes a central transcriptional regulator involved in adhesion, virulence, cell wall regulation, morphogenesis and chlamydospore formation[20,35,38–40]. We observed that *RME1* overexpression under the control of the strong constitutive P_{TDH3} promoter could alleviate the chlamydospore formation defect of the *hog1ΔΔ* mutant (Fig. 6a). In a similar fashion, P_{TET}-driven *RME1* overexpression could restore chlamydosporulation in the *efg1ΔΔ* strain (Fig. 6a). The rounded cells produced by the *hog1ΔΔ* and *efg1ΔΔ* mutants were genuine chlamydospores as they expressed the P_{PGA55}-GFP fusion and showed surface labelling with Csp1-GFP (Fig. 6b). These results indicate that *RME1* overexpression bypasses the Hog1 and Efg1 requirements for chlamydosporulation and suggest that Rme1 acts as a central regulator in this process.

**Sfl1 and Ndt80 are antagonistic regulators of Rme1**. Our previous work suggested that Sfl1, a negative regulator of hyphal growth, positively regulates *RME1* expression (*RME1* is upregulated upon *SFL1* overexpression, and Sfl1 binds to *RME1* regulatory region[22]). On the other hand, Ndt80, a positive regulator of filamentation[41,42], has also been shown to bind to *RME1* promoter region[41] and to repress its expression[42]. Furthermore, *NDT80* expression levels are reduced when *RME1* is overexpressed (Supplementary Data 6). We tested the phenotype of *sfl1ΔΔ* and *ndt80ΔΔ* mutants in both SC5314 and CEC2018 backgrounds, grown in liquid chlamydospore-inducing conditions (Fig. 6c). The *sfl1ΔΔ* mutants produced numerous filaments but no chlamydospores (Fig. 6c), in line with Sfl1 positively regulating *RME1* expression. On the other hand, strains deleted for *NDT80* produced chlamydospores in much greater number than the WT control (Fig. 6c), in agreement with Ndt80 negatively regulating *RME1* expression. In chlamydospore-inducing conditions, overexpression of *SFL1* led to chlamydosporulation, whereas *NDT80* overexpression abolished chlamydospore formation (Fig. 6d), confirming the antagonistic functions of these two transcription factors.

To further explore the relative importance of *SFL1*, *NDT80* and *RME1* in chlamydospore formation, we tested the epistatic relationship between the three regulators in a SC5314 strain-derivative (Supplementary Fig. 6). We overexpressed *RME1* in the *sfl1ΔΔ* strain background and vice versa (Supplementary Fig. 6a). We found that a functional *RME1* gene is required for *SFL1* to induce chlamydosporulation (Supplementary Fig. 6a, upper panel), whereas overexpression of *RME1* does not require *SFL1* to activate chlamydospore formation (Supplementary Fig. 6a, lower panel), indicating that *RME1* acts downstream of *SFL1*. On the other hand, deletion of *RME1* in the *ndt80ΔΔ* genetic background abolished chlamydospore development (Supplementary Fig. 6b), indicating that *NDT80* loss-of-function does not bypass the requirement for *RME1* to produce chlamydospores; further highlighting the central role of *RME1* in this developmental program. These results were confirmed at the transcriptional level by RT-qPCR (Supplementary Fig. 7). Indeed, *RME1*

expression levels decrease when *SFL1* and *NDT80* are respectively deleted and overexpressed (Supplementary Fig. 7). Taken together, our results point to a central role of Rme1 within the transcriptional circuitry that controls chlamydospore development.

**Gene expression levels of other regulators of chlamydosporulation**. We sought to determine if other regulators exhibited differential expression levels in *C. albicans* clinical isolates with different chlamydospore formation efficiency, as shown for *RME1* (Fig. 3). We measured the transcript abundance of another positive regulator of chlamydospore formation, *SFL1*, in both strong (strains CEC3620 and CEC2018) and weak (strains CEC1424 and CEC1426) chlamydospore formers (Supplementary Fig. 8). We similarly found that *SFL1* expression in clinical isolates correlated with their ability to form chlamydospores, being high in strains CEC3620 and CEC2018 (strong chlamydospore formers) and unchanged in CEC1424 and CEC1426 (weak chlamydospore formers, Supplementary Fig. 8, left panel). We extended our analysis to a negative regulator of chlamydospore formation, *NRG1*[25]. We confirmed the phenotype of the *nrg1ΔΔ* mutant (Supplementary Fig. 6c), and showed that deleting *RME1* in the *nrg1ΔΔ* background abolished chlamydospore formation (Supplementary Fig. 6c). Interestingly, *NRG1* displayed an expression pattern that inversely correlated with the efficiency of clinical isolates to form chlamydospores (Supplementary Fig. 8, right panel). *NRG1* transcripts were unaltered or upregulated in weak chlamydospore formers (CEC1424 and CEC1426, Supplementary Fig. 8, right panel) and downregulated in strong chlamydospore formers (CEC3620 and CEC2018). We conclude that the correlation between transcript abundance and the ability to form chlamydospores is not restricted to Rme1 (and its targets), but rather encompasses other regulators of this developmental program, including Sfl1 and Nrg1.

**Discussion**

A hallmark of the human fungal pathogen *Candida albicans* biology is its ability to undergo several morphogenetic programs. It can grow as yeast, pseudohypha or hypha. The yeast form can also switch between a white phase and an opaque phase, and other yeast forms have been described such as the GUT and grey forms (reviewed in ref. [43]). The ability of *C. albicans* to alternate between these morphotypes contributes to its capacity to thrive in a variety of environments and cause disease. Many master regulators of the morphogenetic transitions have been identified, including the transcription factors Efg1 and Wor1[40,44]. In addition, *C. albicans* is able to form chlamydospores that have long been regarded as a diagnostic criterion to distinguish *C. albicans* from other medically relevant yeasts. To date, only a handful of *Saccharomycotina* species have been shown to produce chlamydospores including *C. albicans*, *C. dubliniensis* and, as observed here, the closely related non-pathogenic species *C. buenavistaensis*. Chlamydosporulation in *C. albicans* and *C. dubliniensis* is associated with a specific gene expression pattern, thus constituting another developmental pathway in these species[25]. Yet, the nature of the transcription factor(s) that is(are) central for activating chlamydosporulation in *C. albicans* and related species remained elusive. While characterization of selected *C. albicans* knock-out mutants led to the conclusion that the TOR and cAMP signalling pathways are critical for conveying chlamydospore-inducing nutritional cues[19], there is no evidence that transcription factors in these pathways are direct regulators of the chlamydospore-specific genes. Here, we show that Rme1, a transcription factor shared by most ascomycotina, is central to activation of the chlamydosporulation program: (1) *RME1* is

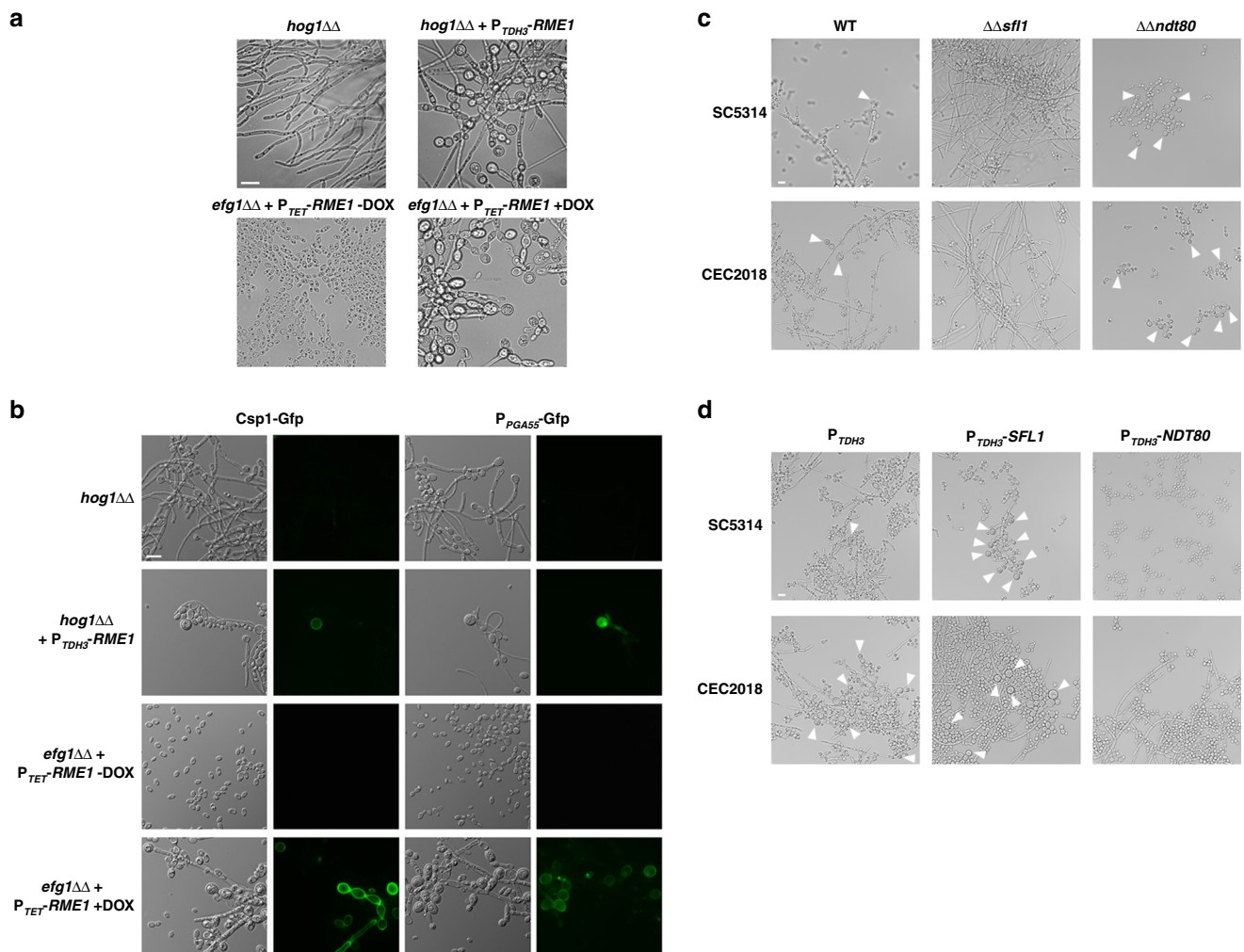

**Fig. 6 Regulators of morphogenesis influence chlamydosporulation through Rme1. a** *C. albicans hog1ΔΔ* and *efg1ΔΔ* mutant strains expressing P$_{TDH3}$-*RME1* and P$_{TET}$-*RME1* respectively were grown during 3 days under chlamydospore-inducing conditions on PCB plates, **b** the same cells recovered from cellophane film on PCB plates expressing either the *CSP1-GFP* (left) or P$_{PGA55}$-*GFP* (right) fusions. The cultures were inspected by phase contrast and fluorescence microscopy. Doxycycline was added when needed to a final concentration of 40 µg/mL. Scale bar = 10 µm. **c** *C. albicans sfl1ΔΔ* and *ndt80ΔΔ* mutant strains in both SC5314 (CEC5288 and CEC5290, respectively, top panels) or CEC2018 (CEC5284 and CEC5286, respectively, bottom panels) backgrounds were grown overnight in liquid chlamydospore-inducing conditions. Scale bar = 10 µm. **d** Strains constitutively overexpressing either *SFL1* (middle panels) or *NDT80* (right panels) in the WT backgrounds of SC5314 (CEC5278 and CEC5280, respectively, top) or CEC2018 (CEC5274 and CEC5276, respectively, bottom) cultured overnight in liquid chlamydospore-inducing conditions; SC5314 and CEC2018 transformed with the empty overexpression plasmid (CEC5393 and CEC5292, respectively) were used as controls (left panels). Strains were examined by light microscopy for their ability to form chlamydospores (white arrowheads in (**c**) and (**d**)). Scale bar = 10 µm.

necessary for chlamydosporulation in *C. albicans, C. dubliniensis* and *C. buenavistaensis*; (2) *RME1* overexpression triggers the formation of chlamydospores in the absence of chlamydospore-inducing cues; (3) *RME1* overexpression can bypass the requirement for other regulators of chlamydosporulation such as Efg1 and Hog1; (4) *RME1* expression levels are highly correlated to the ability of a given *C. albicans* isolate to produce chlamydospores; and (5) Rme1 directly regulates the expression of chlamydospore-specific genes. In agreement with the role of Rme1 as a master regulator of the chlamydosporulation program, we have also shown that *RME1* expression is biphasic. In the first phase, *RME1* expression is activated in response to chlamydospore-inducing environmental cues, independently of the function of Rme1. In a second phase, expression of *RME1* is maintained through a positive feedback loop thus committing *C. albicans* to the formation of chlamydospores even if chlamydospore-inducing cues are removed. This is reminiscent of other sporulation programs

that have been described in fungi such as *Aspergillus* asexual sporulation whereby activation of *brlA* expression (and subsequently *abaA* and *wetA*) commits the fungus to the formation of conidiophores and conidia[45]. Similarly, activation of *IME1* expression is the crux triggering *Saccharomyces cerevisiae* entry into the sporulation pathway[46].

Our results raise the question of the mechanisms by which *RME1* expression is turned on in response to chlamydospore-inducing cues such as low temperature, nutrient restriction and microaerophilia. Several transcription factors, including Sfl1 and Ndt80, have been previously shown to bind the *RME1* promoter region and/or regulate its expression[22,42,47]. Here we show that Sfl1 is a positive regulator of chlamydosporulation while Ndt80 is a negative regulator of this process. Previous results in our laboratory and others have indicated that Sfl1, Ndt80 and Efg1 bind within the same region of several promoters (including that of *RME1*), with Sfl1 negatively regulating hypha-specific genes

while positively regulating yeast-specific genes. Based on our observations, including that the Efg1 requirement for chlamydosporulation is bypassed upon *RME1* overexpression, we propose that chlamydosporulation-inducing cues relieve Ndt80-mediated *RME1* repression, thus allowing Sfl1 and Efg1 to trigger *RME1* activation. Such an activation would only occur at low temperature as upon higher temperature, Sfl1 activity at promoters is counteracted by Sfl2 production and binding to Sfl1-bound promoters[22]. In this respect, chlamydosporulation appears mutually exclusive from true hyphal morphogenesis even though some form of filamentation is associated with the production of suspensor cells and chlamydospores.

*RME1* was originally identified in *Saccharomyces cerevisiae* where it acts as a negative regulator of meiosis through positive activation of the *IRT1* long non-coding RNA that itself negatively regulates the expression of *IME1*, the meiosis master regulator[48]. Notably, Rme1 appears to play roles in various aspects of yeast sexuality although in a manner that varies from species to species. In *Kluyveromyces lactis*, Rme1 acts as a positive regulator of mating-type switching and expression of haploid-specific genes integrating starvation signals[49] while in *Ogataea polymorpha*, Rme1 positively regulates nitrogen-starvation-dependent mating-type switching and mating[50]. Our observation provides yet another function for Rme1, with a role in the formation of asexual chlamydospores, highlighting multiple events of functional rewiring during the evolution of the *Saccharomycotina*. Our comparative analyses of Rme1 homologs from the *Candida* clade as well as those from additional representative species of the ascomycete lineage show low-to-moderate sequence divergence within the DNA-binding domain (DBD, located at the C-terminus), but high sequence divergence in the N-terminal 2/3rds of the proteins from *C. albicans*, *C. dubliniensis*, *C. buenavistaensis* and *C. tropicalis*. Intriguingly, the Rme1 N-termini in *C. albicans*, *C. dubliniensis* and *C. buenavistaensis* carry extensive repeats of poly-Asn/Gln/His stretches that are absent in other *Candida* species. More precisely, while *C. albicans* Rme1 DBD displays high sequence similarity with the equivalent sequences from the phylogenetically related species *C. dubliniensis*, *C. buenavistaensis*, *C. tropicalis*, *L. elongisporus*, *C. orthopsilosis*, *C. parapsilosis* and *C. metapsilosis*, sequence similarity drastically decreases in the N-terminal 2/3rds of the proteins when we compare Rme1 sequences from *C. albicans*, *C. dubliniensis*, *C. buenavistaensis* and *C. tropicalis* to those from *L. elongisporus*, *C. orthopsilosis*, *C. parapsilosis* and *C. metapsilosis*. This suggests that species-specific differences in the Rme1 protein underlie its role in the regulation of chlamydospore formation and that functional repurposing might have occurred—at least in part—through substitutions affecting the N-terminal half of the protein, which may include interaction domains with additional (co-)regulators/regulatory proteins and/or the RNA-polymerase II holoenzyme. We have also performed motif discovery analyses to determine if *C. albicans* Rme1 binds to a divergent *cis*-acting element. We did not identify regular patterns that could be indicative of motif conservation and/or overrepresentation in *C. albicans* Rme1 target promoters. Alternatively, a search of the motif identified in *S. cerevisiae* (5′-GWACCWCAADA-3′) did not return any significant matches/hits. We think that functional repurposing might have occurred through alteration of both Rme1 functional domains and *cis*-acting elements. Our future work will be complemented by extensive bioinformatic analyses to delineate the mechanistic aspects that potentially led to neofunctionalization of *C. albicans* Rme1.

In *C. albicans*, Rme1 offers another example of functional reassignment of genes implicated in meiosis, a process that has been lost in this and closely related species. Interestingly, these genes have been reassigned to the control of various differentiation pathways. While *UME6*, a positive regulator of meiosis in *S. cerevisiae*, positively regulates hyphal morphogenesis in *C. albicans*, *RME1*, a negative regulator of meiosis in *S. cerevisiae*, has been rewired to positively regulate chlamydosporulation, which, as discussed above, appears antagonistic with true hyphal morphogenesis. The involvement of *RME1* in chlamydosporulation appears to be recent as this process is restricted to *C. albicans* and its close relatives *C. dubliniensis* and *C. buenavistaensis*, and is lacking in *C. tropicalis* and *C. parapsilosis*. Additional evidence for a recent specialization lies in the observation that *C. tropicalis* or *C. parapsilosis* *RME1* cannot replace the function of *C. albicans* *RME1*. Moreover, genes associated with chlamydosporulation show specificity to *C. albicans* and *C. dubliniensis*. This is in particular the case for the *CSP* gene family[25] whose expansion is restricted to chlamydosporulating species. The function of chlamydospores in *C. albicans* biology remains to be understood. The fact that this differentiation occurs when *C. albicans* is exposed to unusual growth conditions such as poor nutrient availability, low temperature, darkness and microaerophilia could suggest that chlamydospores arise in very specific body niches or in the environment. Identification of the central role of Rme1 in chlamydosporulation control may bring forth the underlying molecular mechanisms, and unveil the biological function of chlamydospores.

## Methods

**Strains, media and growth conditions**. The *C. albicans* strains used in this study are listed in Supplementary Data 1. Depending on the experimental conditions, the strains were either grown at 30 °C in YPD medium [1% (w/v) yeast extract, 2% (w/v) peptone and 2% (w/v) dextrose], SC medium [synthetic complete, 0,67% (w/v) yeast nitrogen base without amino acids (YNB, Difco®), 2% (w/v) dextrose, 0.077% (w/v) complete supplement mixture minus uracil)] or SD medium [synthetic dextrose, 0,67% (w/v) yeast nitrogen base with 2% (w/v) glucose, supplemented if necessary with arginine, histidine or uridine (20 mg/L each)]. Chlamydospore formation was achieved by growing *C. albicans* strains at 25 °C, in darkness, either in liquid PCB medium[51] [10% potato/carrot extract and 1.5% dehydrated fresh bile (Difco®)] during 24 h with gentle agitation (70 rpm) or for 3 days on PCB agar covered with a cellophane sheet (Dutscher) to prevent agar invasion. Tetracycline-inducible promoter expression ($P_{TET}$) was achieved by adding 40 μg/mL of doxycycline (Sigma®) to the medium. For all media, solid plates were prepared with 2% (w/v) agar.

**Chlamydospore staining**. SC5314, *rme1*ΔΔ and $P_{TET}$–*RME1* overexpression strains were grown in liquid PCB medium at 25 °C, in darkness, for 48 h, adding 40 μg/mL doxycycline for $P_{TET}$ induction. Cells were rinsed with McIlvaine's buffer (0.2 M Na$_2$HPO$_4$/0.1 M citric acid pH 6.0)[29] or 1× PBS and incubated in the dark with either 10 μg/mL Eosin Y for 1 h, or with 2 μg/mL BODIPY for 30 min[52], respectively; cells were rinsed twice with either McIlvaine's buffer or 1× PBS prior to fluorescent microscopic observation.

**Microscopy**. Aliquots of liquid cultures were centrifuged, the pellets rinsed in water, and cells transferred on microscope slides, and observed either using a Leica DM RXA microscope (Leica Microsystems) equipped with an oil-immersed 100× objective, 1.4 N.A. (images were captured with a Hamamatsu ORCA II-ER cooled CCD camera, using Openlab software version 3.5.1, Improvision Inc.) or with an Olympus IX 83 microscope equipped with either a 100× oil-immersed objective or a 40× objective (images were captured with a Hamamatsu ORCA Flash cooled CCD camera, using Cell sens software). Colonies growing on PCB plates were observed with a Leica DM RXA microscope (Leica Microsystems) equipped with an oil-immersed 40× objective, and images captured as above. Images were then processed with Adobe Photoshop CS6.

**Candida transformation**. *C. albicans* and *C. buenavistaensis* transformations were performed using the lithium-acetate transformation protocol developed by Walther and Wendland[53]. In short, cells were diluted to OD$_{600}$ = 0.2 in 50 mL YPD and grown at 30 °C until OD$_{600}$ = 0.6–0.8. Cells were pelleted, rinsed first in ice-cold 10× TE (100 mM Tris pH7.5/10 mM EDTA) then in ice-cold 1× TE/100 mM lithium acetate, and resuspended in 200 μL 1× TE/100 mM lithium acetate. After 1 h incubation on ice, 50 μL of competent cells were mixed with up to 1 μg DNA, 50 μg of salmon-sperm DNA, and 300 μL of a 40% PEG 3000/1× TE/100 mM lithium acetate solution. The transformation mix was incubated O/N at 30 °C; cells were then heat-shocked at 44 °C for 15 min and washed once with 500 μL of SD medium; after centrifugation, the pellet was resuspended either in 300 μL of SD

medium and plated on selection plates for prototrophic selection, or in 1 mL YPD, the cells incubated at 30 °C for 3–4 h before plating on drug-containing media. *C. buenavistaensis* overnight cultures were incubated at 25 °C to prevent filamentation. *C. dubliniensis* was transformed by electroporation, as described[54]. In brief, the cells were grown to $OD_{600} = 1.6–2.2$. Cells were centrifuged, resuspended in 10 mL of 1× TE/100 mM lithium acetate and incubated for 1 hr at 30 °C; after adding 250 µL 1 M DTT, cells were kept at 30 °C for 30 min. Cells were rinsed using large volumes of ice-cold water, then ice-cold 1 M sorbitol, and resuspended in 50 µL of ice-cold 1 M sorbitol. Forty microlitres of competent cells were mixed with 1 µg DNA, and electroporation was performed at 1.6 kV, 200 Ω, 25 µF, in a 0.2 cm cuvette. Cells were resuspended in YPD. The selection of the transformants was based on either prototrophy for uridine, histidine or arginine[55], or nourseo-thricin-[56] or hygromycin B-resistance[57].

**Construction of C. albicans RME1 knockout and overexpression strains.** The *RME1* deletion mutant was generated in the BWP17 background by knocking out both alleles by cassette replacement. In brief, the *HIS1* and *ARG4* disruption cassettes were generated by PCR using pSN52-CdHIS1 and pSN69-CdARG4[55] as templates, respectively, and specific primers containing ca. 100 bp homolog to the *RME1* gene (RME1-KO-F and RME1-KO-R; all primer sequences are listed in Supplementary Data 2). Proper gene replacements were verified by PCR with RME1-KO-DETECT-F, RME1-KO-DETECT-R, RME1-500-F and RME1-500-R, CdHIS1-DETECT-F, CdHIS1-DETECT-R, CdARG4-DETECT-F and CdARG4-DETECT-R. The resulting *ura*- auxotrophic clones were transformed either with CIp10[58] to generate prototrophic deletion mutants or with pNIMX[59] to generate strains expressing the *C. albicans*-adapted reverse Tet-dependent transactivator (*cartTA*). Gene overexpression was achieved using the Gateway® technology (Invitrogen®). In short, the *RME1* ORF borne on pDONR207 was transferred using the Invitrogen® Gateway LR clonase to the destination vectors CIp10-$P_{TET}$-GTW[59], CIp10-$P_{TET}$-rTAP-GTW, and CIp10-$P_{TET}$-GFP-GTW[60]. In addition, the *RME1* genes from *C. dubliniensis*, *C. buenavistaensis*, *C. tropicalis* and *C. parapsilosis* were PCR-amplified on genomic DNAs with specific primers containing the *attB1* and *attB2* sites (Cd-GTW-RME1fwd, Cd-GTW-RME1rev, Cb-GTW-RME1fwd, Cb-GTW-RME1rev, Ct-GTW-RME1fwd, Ct-GTW-RME1rev, Cp-GTW-RME1fwd and Cp-GTW-RME1rev, Supplementary Data 2), which allowed cloning the ORFs by a recombination-mediated transfer into the Gateway™ donor vector pDONR207 using the Invitrogen® Gateway BP clonase, and then to CIp10-$P_{TET}$-GTW and CIp10-$P_{TET}$-rTAP-GTW[59] with the LR clonase. The resulting plasmids were StuI-digested prior to transforming *C. albicans* expressing the *cartTA* transactivator. Clones were verified by PCR with primers CIp_UL and CIp_UR[59]. *C. buenavistaensis RME1* sequence has been deposited in GenBank under accession number MK070497.

**Murine candidemia model.** Female BALB/c mice at 6–8 weeks of age were used for disseminated candidemia following intravenous administration. Animals were separated according to their strain and housed in top-filtered cages with sterile water and food provided ad libitum. Throughout the study, mice were kept on 12 h light/dark cycles, with lights on from 07:00 h to 19:00 h, at 20 °C with ambient humidity. Mice were infected intravenously with $1 \times 10^6$ CFU of WT or *rme1ΔΔ C. albicans* strains (CEC1, CEC4788, CEC4783, CEC4785 and CEC4805) in a 0.1 mL volume via the tail vein. Control mice received sterile 9°/°° NaCl. For survey experiments, mice were monitored for more than 3 weeks after infection. Body weight was monitored twice per week. Moribund mice were humanely sacrificed, and their death recorded as occurring on the following day. All procedures were approved by a local ethics committee and the French Ministry of Higher Education, Research and Innovation (Authorization N°8384) in accordance with the European Communities Council Directive (86/609/EU) and the European Union guidelines.

**Chromatin immunoprecipitation and ChIP-on-chip assay.** The *rme1ΔΔ* mutant strain expressing the tandem affinity purification epitope (TAP)-*N*-tagged ($P_{TET}$-rTAP-*RME1*) or the untagged ($P_{TET}$-*RME1*) versions of Rme1 were grown overnight in YPD medium at 30 °C, then diluted to an $OD_{600}$ of 0.2 in 50 mL of YPD medium supplemented with 40 µg/mL doxycycline and grown for 4 h at 30 °C. The cultures were treated with 1% formaldehyde and snap-frozen in liquid nitrogen. Cells were lysed with glass beads using a FastPrep-24 instrument (MP Biomedicals®) with 10 cycles of 40 s each at 5.5 m/s with a 1 min on ice in between. The soluble chromatin fragments were obtained by sonication, six times during 20 s each at power 8 for an output signal amplitude of 15 (Microns, Peak to Peak) using a probe sonicator (MSE), yielding DNA fragments with an average size ~200–500 bp. The immunoprecipitation was performed overnight at 4 °C with 500 µL of clarified sonicated extracts and 40 µL of IgG-coated magnetic beads (Dynabeads Pan mouse IgG, Invitrogen®), previously pre-hybridized overnight with PBS-0.1% BSA at 4 °C. DNA was labelled as follows. In brief, the immunoprecipitated (IP) DNA fragments were blunted with T4 DNA polymerase and ligated to unidirectional linkers. The DNA was amplified by ligation-mediated PCR in the presence of aminoallyl-modified dUTP. The labelling was carried out post-PCR using mono-reactive Cy dye N-hydroxysuccinimide esters (Cy5/Cy3 monoreactive dye packs; Amersham Biosciences®) that react specifically with the aminoallyl-modified dUTP [5-(3-aminoallyl)-2′ deoxyuridine-5′triphosphate; Sigma-Aldrich®]. The labelled IP

DNA from the tagged and untagged strains were mixed; 25 µL of 2× HI-RPM Hybridization Buffer and 5 µL of 10× CGH Blocking agent were added in a total volume of 50 µL and the mix was denatured for 3 min at 95 °C then kept at 42 °C before hybridization to the *C. albicans* whole-genome tiled DNA microarrays from Assembly 20 (GEO platform accession GPL13696, https://www.ncbi.nlm.nih.gov/geo/query/acc.cgi?acc=GPL13696) overnight at 65 °C. Slides were washed in Agilent wash buffers 1 (at room temperature for 1 min) and 2 (at 37 °C for 1 min) then removed slowly from the washing dish to minimize droplets on the slide. Scanning of the arrays was performed using an Axon Autoloader 4200AL scanner and analyzed with the GenePix software version 7.0 (Molecular Devices). Data normalization (Quantile) and peak finding were conducted using CisGenome[24]. In order to visualize the ChIP-on-chip results we used the Integrated Genomics Viewer (IGV) software[61]. The complete Rme1 binding datasets are available in Supplementary Data 3–5.

**Whole-genome transcript profiling analyses.** For the $P_{TET}$-*RME1* microarray experiment, three independently generated *rme1ΔΔ* mutant strains expressing $P_{TET}$-*RME1* (see Supplementary Data 1) were grown overnight at 30 °C in YPD medium, then diluted in fresh YPD medium to an $OD_{600}$ of 0.2 and grown during 2 and 4 h (supplemented or not with 40 µg/mL doxycycline). For comparing the transcriptomes of CEC2018 and its relative CEC2018-*rme1ΔΔ* the strains grown overnight at 30 °C in YPD medium, then diluted to an $OD_{600}$ of 0.2 in 10 mL of PCB liquid medium and cultured at 25 °C in the dark and under microaerophilic conditions during 24 h. For both experiments, total RNA was isolated using the hot phenol protocol. In brief, cells were resuspended in 375 µL TES buffer (10 mM Tris pH 7.5, 10 mM EDTA, 0.5% SDS) at room temperature, after which 375 µL acid Phenol:Chloroform (5:1, Amresco, Solon, OH) were added. Samples were then incubated for 1 h at 65 °C with vigorous vortexing during 20 s each 10 min, and then centrifuged for 20 min at 14,000 rpm. The supernatants were mixed with 750 µL acid Phenol:Chloro-form (5:1) and centrifuged at 14,000 rpm for 10 min. The aqueous phase was mixed with 750 µL Chloroform:Isoamyl alcohol (24:1, Interchim, Montluçon, France) and centrifuged at 14,000 rpm during 10 min. The aqueous layer was mixed with 1 mL 99% ethanol (pre-cooled at −20 °C) and 37 µL of 3 M sodium acetate pH 5.0, and centrifuged at 14,000 rpm for 40 min at 4 °C. The pellet was washed in 500 µL 70% ethanol, and the RNA was collected by centrifugation at 14,000 rpm for 20 min at 4 °C. The RNA pellet was resuspended in 150 to 300 µL RNase-free water. The RNA was stored at −80 °C until needed. First-strand cDNA was synthesized from 20 µg of RNA and labelled with Cy5 (doxycycline-induced cDNA samples for the $P_{TET}$-*RME1* experiment and CEC2018-*rme1ΔΔ* for the CEC2018-*rme1ΔΔ* experiment) and Cy3 (non-induced cDNA samples for the $P_{TET}$-*RME1* experiment and CEC2018-WT for the CEC2018-*rme1ΔΔ* experiment) using the Superscript III indirect cDNA labelling system (Invitrogen®). The samples were purified, resuspended in 10 µL of DNase-free water, mixed and 2.5 µL of 10× blocking agent and 12.5 µL of 2× hybridization buffer (Agilent Technologies®) were added. The samples were denatured 3 min at 95 °C prior to hybridization to a *C. albicans* expression microarray (Agilent Technologies®, https://www.ncbi.nlm.nih.gov/geo/query/acc.cgi?acc=GPL19289) designed to cover the whole *C. albicans* genome, carrying two non-overlapping probes for each of the 6,105 ORFs from Assembly 19, for a total of 15,744 probes. The hybridization was performed as described for the ChIP-on-chip experiment. Scanning of the expression arrays was done using an Axon Autoloader 4200AL scanner (Molecular Devices, Downington, PA) and images were analysed with the GenePix Pro 6.1.0.2 software (Molecular Devices, Downington, PA). The GenePix files (.gpr) were imported into the Arraypipe 2.0 software for spot filtering, background subtraction (limma normexp BG correction) and Lowess global normalization of signal intensities[62]. Replicate arrays ($n = 3$) were combined and fold-change and *P*-values (Welch's *t*-test within group) were calculated. The complete expression profiling datasets are available in Supplementary Data 6 and 8. As a control, we also assessed the effect of doxycycline on gene transcription[26].

**Validation of transcriptomics and ChIP-on-chip analyses.** Total RNA from strains expressing $P_{TET}$-*RME1* (see Supplementary Data 1) or from CEC2018 WT and *rme1ΔΔ* strains was isolated as described before, and used to perform a reverse transcription (RT) reaction (2 µg of total RNA) using the SuperScript III enzyme (Invitrogen®) in a final reaction volume of 20 µL. The qPCR reactions contained the following components: 1 µL of cDNA from the RT reaction, 0.5 µL of each primer (forward and reverse of a given gene, Supplementary Data 2) at 10 pmol/µL, 7.5 µL of 2× Takyon Rox SYBR MasterMix dTTP blue (Eurogentec®) and 5.5 µL $H_2O$. The reactions were performed in a MicroAmp Optical 96-Well Reaction Plate (Applied Biosystems®) using an Eppendorf Realplex Mastercycler real-time PCR instrument (Eppendorf®) with 1 cycle at 50 °C for 2 min, 1 cycle at 95 °C for 10 min and 50 cycles at 95 °C for 15 s and 58 °C for 1 min. The results obtained were analysed in the *realplex* software version 2.2 (Eppendorf®), where the threshold cycle ($C_T$) values were determined. The *TEF3* and *ACT1* genes were used as calibrator and as a gene control respectively [using the primers TEF3-F and TEF3-R, or ACT1-F and ACT1-R, respectively (Supplementary Data 2)]. For the overexpression strains, the relative gene expression at 2 h and 4 h (n-fold, for the samples induced with doxycycline as compared with the non-induced controls) for the following genes: *ORF19.654* (ORF19.654_F_qPCR and ORF19.654_R_qPCR), *CSP1* (CSP1_F_qPCR and CSP1_R_qPCR), *CSP2* (CSP2_F_qPCR and CSP2_R_qPCR), *PGA55*

(PGA55_F_qPCR and PGA55_R_qPCR), FEN1 (FEN1_F_qPCR and FEN1_R_qPCR) and ORF19.7250 (ORF19.7250_F_qPCR and ORF19.7250_R_qPCR), was validated by using the $2^{-\Delta\Delta CT}$ method, in the following manner: $\Delta C_T = C_T$ (selected gene) $- C_T$ (TEF3 calibrator gene) calculated for both doxycycline-induced and uninduced samples, and $\Delta\Delta C_T = \Delta C_T$ (doxycycline-induced sample) $- \Delta C_T$ (uninduced sample). For the CEC2018 rme1ΔΔ strain, the relative gene expression as compared to the WT was determined, as before, for the following genes: ORF19.654, CSP1, CSP2 (primers as before), ORF19.4463 (ORF19.4463_F_qPCR and ORF19.4463_R_qPCR), ORF19.555 (ORF19.555_F_qPCR and ORF19.555_R_qPCR), ORF19.6660 (ORF19.6660_F_qPCR and ORF19.6660_R_qPCR), GAL7 (GAL7_F_qPCR and GAL7_R_qPCR) and MAC1 (MAC1_F_qPCR and MAC1_R_qPCR). For the ChIP-on-chip experiment, the target-DNA enrichment (n-fold, primer sequences are shown in Supplementary Data 2) for ORF19.654 (ORF19.654_F_CHIPPCR and ORF19.654_R_CHIPPCR), CSP1 (CSP1_F_CHIPPCR and CSP1_R_CHIPPCR), CSP2 (CSP2_F_CHIPPCR and CSP2_R_CHIPPCR), PGA55 (PGA55_F_CHIPPCR and PGA55_R_CHIPPCR), FEN1 (FEN1_F_CHIPPCR and FEN1_R_CHIPPCR), ORF19.7250 (ORF19.7250_F_CHIPPCR and ORF19.7250_R_CHIPPCR) and ACT1 (ACT1-F and ACT1-R, used as a negative control) were calculated using the relative quantification by using the $2^{-\Delta\Delta CT}$ method, as follows: $\Delta C_T = C_T$ (target) $- C_T$ (TEF3 calibrator gene) calculated in both tagged (TAP-Rme1 strain CEC4745) and untagged (untagged Rme1, strain CEC4741) samples, and $\Delta\Delta C_T = \Delta C_T$ (tagged) $- \Delta C_T$ (untagged), where $C_T$ (TEF3 reference) is the $C_T$ for the TEF3 amplicon (primers TEF3-F and TEF3-R). Assays were performed in triplicate using three biological replicates for the RT-qPCR experiments and three times independently with two biological replicates for the ChIP-qPCR experiments. Data were expressed as the mean ± SD. Differences for a given gene/ChIP target were analysed by the two-tailed unpaired Student's t-test, the n-fold relative gene expression values to the corresponding n-fold values of ACT1 control, or for the ChIP-qPCR the n-fold enrichment values of the target gene (ORF19.654, CSP1, CSP2, PGA55, FEN1 and ORF19.7250) to those of the corresponding ACT1 control, a P value < 0.05 was considered as statistically significant.

**Plasmid constructions for the CSP1- and P_PGA55-GFP fusions.** The CSP1-GFP fusion was generated as follows: first, the CSP1 gene was PCR amplified and cloned into pDONR207, and then transferred into the destination vector CIp10-P_ACT1-GTW-GFP[60] as described above, yielding CIp10-P_ACT1-CSP1-GFP. This plasmid was subsequently digested with EcoRI and self-ligated to remove the URA3 marker. The resulting plasmid was BglII/EcoRV-digested in order to substitute the ACT1 promoter with a BamHI/EcoRV fragment from CSP1 promoter region (obtained by PCR with primers F-Upstream-CSP1 and R-Upstream-CSP1) yielding CIp10-P_CSP1-CSP1-GFP. The CaHygB gene was obtained by digesting the plasmid pAU34-CaHygB[57] with BglI and XbaI, and cloned into CIp10-P_CSP1-CSP1-GFP digested with NheI and SmaI. Finally, the resulting plasmid CIp10-P_CSP1-CSP1-GFP-HygB was AleI-digested and used to clone CSP1 downstream region (PCR amplified with primers Fwd-Downstream-CSP1 and Rev-Downstream-CSP1), obtaining the plasmid designated CIp10-P_CSP1-CSP1-GFP-HygB-T_CSP1. This plasmid was digested with AflII and AleI prior to C. albicans transformation. The proper integration was verified by PCR by using the primers CPS1-GFP-verif-F, CSP1-GFP-verif-R, Hygro-verif-F and GFP-verif-R. To construct the P_PGA55-GFP reporter fusion, CIp10-P_CSP1-CSP1-GFP-HygB was digested with PstI and Acc65I in order to place PGA55 upstream region (obtained by PCR with primers F-PrPGA55 and R-PrPGA55 and cut with PstI/SpeI) upstream of GFP. The resulting plasmid, CIp10-P_PGA55-GFP-HygB, was subsequently digested with AleI and BamHI to clone the PGA55 terminator region (amplified by PCR with primers F-TerPGA55 and R-TerPGA55 and cut by AleI/BamHI), resulting in the final construct CIp10-P_PGA55-GFP-HygB-T_PGA55; this plasmid was digested with ScaI and DraIII prior to transforming C. albicans. Integration was verified by PCR with primers PGA55-GFP_verif_F, PGA55-GFP_verif_R, Hygro-verif-F and GFP-verif-R.

**Plasmid construction for the P_RME1-gLUC fusion.** The plasmid CIp10-P_ACT1-gLUC59[63] was XhoI/HindIII-double digested and used to clone a 530 bp fragment corresponding to the RME1 upstream region to create the plasmid CIp10-P_RME1-gLUC59, which was subsequently EcoRI/NheI-double digested, blunted with Klenow DNA polymerase and used to clone the CaHygB gene, obtained by the digestion of the plasmid CIp10-P_CSP1-CSP1-GFP-HygB with DraI. The resulting plasmid CIp10-P_RME1-gLUC59-HygB was digested with NotI and used to clone a 536 bp fragment belonging to the RME1 terminator region to obtain the final construct CIp10-P_RME1-gLUC59-HygB-T_RME1. The plasmid was cut with AleI and KpnI prior to transforming C. albicans. Proper integration at the RME1 locus was verified by PCR with primers Rme1-verify-locus-F, Rme1-verify-locus-R, gLUC-verify-R and CaHygB-detect2-R.

**Building of the C. albicans CEC2018 and C. dubliniensis rme1ΔΔ mutants.** The RME1 coding region was disrupted in the C. albicans clinical isolate CEC2018 and in C. dubliniensis using the SAT1 flipping method developed by Reuß et al.[64]. In order to generate the cassette to knockout CaRME1, a 543 bp fragment in the RME1 upstream region was amplified by PCR with primers RME1-KO-flip-F-Prom and RME1-KO-flip-R-Prom (Supplementary Data 2); the PCR product was then digested with KpnI and XhoI and cloned into the same sites of pSFS1[64] to generate pSFS1-P_RME1. Independently, a 515 bp fragment belonging to the RME1

downstream region was amplified by PCR using the primers RME1-KO-flip-F-Ter and RME1-KO-flip-R-Ter, then digested with NotI and SacI and cloned into the same sites of pSFS1-P_RME1 to generate the disruption plasmid pSFS1-P_RME1-T_RME1, that was linearized with KpnI and SacI prior to transforming the CEC2018 strain. Transformants were selected on YPD plates supplemented with 200 μg/mL of nourseothricin (clonNAT, Werner BioAgents®) after 2 days at 30 °C. The disruption of a single allele of RME1 was confirmed by PCR using the primers RME1-FLIP-VERIFY-KO-F, SAP2-VERIFY-R, PSFS1-VERIFY-F and RME1-FLIP-VERIFY-KO-R (Supplementary Data 2). Two independent transformants were grown overnight in YCB-BSA medium [(2.34% (w/v) yeast carbon base and 0.4% BSA)] at 30 °C to induce the loss of the SAT1 cassette via intramolecular recombination at the FRT sites by the site-specific recombinase FLP and to regenerate the sensitivity to nourseothricin. This event was verified by PCR using the primers RME1-FLIP-VERIFY-KO-F and RME1-FLIP-VERIFY-KO-R, yielding a 2 kbp fragment confirming the excision of the SAT1-FLP cassette. The disruption of the second allele of RME1 was achieved similarly and confirmed by PCR using the primers RME1-500-F and RME1-500-R. To delete the C. dubliniensis ortholog of RME1 (Cd36_06830; CdRME1), a disruption cassette was generated by PCR amplification of the SAT1-flipper cassette from the pSFS2A plasmid[64] using primers with homology to the 5′ and 3′ flanking regions of CdRME1 (CdRMEΔF and CdRMEΔR). The deletion cassette was used to transform C. dubliniensis Wü284 and delete both copies of CdRME1 successively[65]. Deletion of both RME1 alleles was confirmed by PCR with primers flanking the disruption site (CdRMEUP and CdRMEDN) and internal to the disrupted ORF (CdRMEINTF and CdRMEINTR).

**Construction of knockouts with the transient CRISPR-Cas9 system.** Deletions of SFL1 and NDT80 in strains SC5314 and CEC2018, and of the RME1 ortholog in C. buenavistaensis were performed using the transient CRISPR-Cas9 system[66] with modifications. First, nested PCR was used to generate sgRNA cassettes for each ORF. Plasmid pV1093[67] served as template DNA and primer SNR52/F[66] was paired with primers SNR52/R_SFL1, SNR52/R_NDT80 or SNR52/R_RME1_Cb to amplify the SNR52 promoter sequence fragment of each cassette. The second fragment encoding the sgRNA scaffold sequence was amplified with primer sgRNA/R[66] paired with sgRNA/F_SFL1, sgRNA/F_NDT80 or sgRNA/F_RME1_Cb. Primer extensions were used to introduce the specific PAM motif sequence (20 bp) for each corresponding ORF. After hybridization of the two fragments, primers SNR52/N and sgRNA/N[66] were used to amplify the final sgRNA expression cassettes. The resulting cassettes were ligated into TOPO-TA vectors (Invitrogen) and sequence integrity was confirmed by sequencing. Second, repair-template encoding fragments were amplified from plasmid pV1090[67]. Primers SFL1_repair/F with SFL1_repair/R, primers NDT80_repair/F with NDT80_repair/R and primers RME1_Cb_repair/F with RME1_Cb_repair/R amplified the nourseothricine resistance cassette encoding sequence (SAT1) adding 5′- and 3′-homologous sequences of the corresponding ORF to the amplicon, allowing insertion of the cassette by homologous recombination. A sequence encoding the CAS9 cassette was amplified from plasmid pV1093[67] with primers CaCas9/R and CaCas9/F[66] and co-transformed with the sgRNA encoding cassettes and the corresponding repair template-encoding fragment in a 1:1:1 ratio (1 μg each). Homozygous mutants were confirmed by PCR with primers ColoORF_SFL1/F with ColoORF_SFL1/R, ColoORF_NDT80/F with ColoORF_NDT80/R and ColoORF_RME1_Cb/F with ColoORF_RME1_Cb/R internal to the disrupted ORF. Removal of the sgRNA expression cassette and the CAS9 expression cassette was validated by analytic PCR with primer pairs sgRNA/F and sgRNA/R or pV1093_seq_3F and pV1093_seq_4R, respectively, according to Min et al.[66]. The transient CRISPR-Cas9 system was also used to construct double mutants of NDT80 or NRG1 in the rme1ΔΔ mutant strain CEC5914, deriving from SN76. In brief, ARG4 disruption cassettes were generated by PCR using pSN69-CdARG4 as template, and primers NDT80_SN_repair/F with NDT80_SN_repair/R and NRG1_SN_repair/F with NRG1_SN_repair/R. For disruption of NDT80 the generated sgRNA expression cassette was reused (as described above) while for disruption of NRG1 a new sgRNA expression cassette was generated with primers sgRNA/R paired with sgRNA/F_NRG1 and SNR52/F paired with SNR52/R_NRG1. The sgRNA cassettes were PCR amplified and co-transformed with the corresponding repair template fragment and a sequence encoding the CAS9 expression cassette. Primers ColoORF_NDT80/F with ColoORF_NDT80/R and ColoORF_NRG1/F with ColoORF_NRG1/R were used to confirm homozygous mutants.

**Chlamydospore formation screening in C. albicans clinical isolates.** C. albicans clinical isolates (Supplementary Data 7), the WT reference strain SC5314 and the RME1 overexpression strain CEC4297 were grown for 24 h in 3 mL of PCB liquid medium at 25 °C in 24 deep well plates in the dark and under microaerophilic conditions. Cells were centrifuged and washed twice with water and photographed at the microscope. Chlamydospore formation scores were based on the amount of chlamydospores found by field; score 0 was given to strains unable to form chlamydospores under the tested conditions. Score 1 was attributed when a single chlamydospore was observed per field and score 5 was assigned to the isolates that formed masses of chlamydospores. Scores ranging from 2 to 4 were attributed to the isolates that showed intermediate phenotype, depending on the amount of

chlamydospores found per field. The score obtained by each isolate reflects the average of three independent experiments.

**Expression levels of *RME1* and its targets in clinical isolates**. The *C. albicans* wild-type strain SC5314 and the clinical isolates CEC3620, CEC3683, CEC2018, CEC4039, CEC1424, CEC1426 and CEC3539 were used to inoculate 10 mL of PCB in 50 mL flasks sealed with parafilm, and incubated in the dark at 25 °C under gentle agitation (rpm) for 24 h. Total RNA was extracted using the hot phenol protocol and used in RT-qPCR reactions as described before. The results were analysed in the *realplex* software version 2.2 (Eppendorf®), where the threshold cycle ($C_T$) values were determined. The *ACT1* and *MAC1* genes were used as calibrator and gene control respectively [using the primers ACT1-F and ACT1-R or MAC1_F_qPCR and MAC1_R_qPCR, respectively (Supplementary Data 2)]. The relative gene expression (n-fold, for the clinical isolates as compared with the reference strain SC5314) for the following genes: *RME1* (RME1_F_qPCR and RME1_R_qPCR), *PGA55* (PGA55_F_qPCR and PGA55_R_qPCR), *CSP1* (CSP1_F_qPCR and CSP1_R_qPCR), *CSP2* (CSP2_F_qPCR and CSP2_R_qPCR) and *ORF19.654* (ORF19.654_F_qPCR and ORF19.654_R_qPCR) was validated by using the $2^{-\Delta\Delta CT}$ method, in the following manner: $\Delta C_T = C_T$ (selected gene)$-C_T$ (*ACT1* calibrator gene) calculated for both reference strain (SC5314) and clinical isolates, and $\Delta\Delta C_T = \Delta C_T$ (clinical isolates)$-\Delta C_T$ (reference strain). Similarly, relative gene expression levels of *SFL1* (SFL1_F_qPCR and SFL1_R_qPCR) and *NRG1* (NRG1_F_qPCR and NRG1_R_qPCR) were measured in strains SC5314, CEC3620, CEC2018, CEC1424 and CEC1426. For all strains, three technical replicates were performed for three biological replicates. Data were expressed as the mean ± SD. Differences were analysed by the two-tailed Student's *t*-test, a *P* value < 0.05 was considered as statistically significant.

**Autoregulation assay**. The *C. albicans* rme1ΔΔ knockout mutant and WT strains expressing both $P_{TET}$-*RME1* and the *CSP1-GFP* fusion were incubated overnight in YPD liquid medium at 30 °C, then diluted to an $OD_{600}$ of 0.2 in 25 mL of YPD liquid medium supplemented with 40 μg/mL doxycycline and grown for 3 h at 30 °C. Cells were harvested by centrifugation, washed with water and resuspended in fresh YPD liquid medium without doxycycline and incubated during 21 h at 30 °C. Cells were inspected by phase contrast and fluorescence microscopy.

**Luciferase assay**. *C. albicans* strains were grown overnight in YPD liquid medium at 30 °C, and then diluted to an $OD_{600}$ of 0.2 in 12 mL of PCB liquid medium. Cultures were incubated at 25 °C in the dark, and after 10 h of growth, cells were harvested by centrifugation, washed with water and resuspended in 12 mL of liquid YPD medium and incubated up to 24 h. Aliquots of the culture were taken at every time point. Cells were collected by centrifugation washed and resuspended in 100 μL of luciferase assay buffer[68] [0.5 NaCl, 0.1 M $K_2HPO_4$ (pH 6.7), 0.1 mM EDTA and protease inhibitor cocktail (Roche®)] and transferred to a microtiter black transparent 96-well plate. 100 μL of a 2 μM coelenterazine solution (Invitrogen®) were added to each well, mixed with the cells and luminescence was recorded immediately in a microtiter plate reader (Infinite M200, Tecan), with an integration time of 1000 ms.

**Protein preparation and western blotting**. Total protein extracts were obtained from 24 $OD_{600}$ units of a *C. albicans* rme1ΔΔ strain expressing $P_{TET}$-rTAP-*RME1*, with or without the addition of 40 μg/mL doxycycline for inducing the $P_{TET}$ promoter. Overnight cultures centrifuged at 3500 rpm during 5 min at room temperature and the pellets were resuspended in Lysis buffer (50 mM Tris pH 7.5, 1.5 mM EDTA, 140 mM NaCl, 1% Triton X100, 0.1% Na-deoxycholate) with protease inhibitor cocktail (Roche®) and 1.5 mM phenylmethylsulfonyl fluoride (PMSF). The equivalent of 100 μL of glass beads was added to the tubes and then the suspensions were lysed in the FastPrep-24 (MP Biomedicals®), six times during 40 s with 1-min incubations on ice in between. The lysates were centrifuged 10 min at 13,000 rpm. Supernatants were boiled for 1 min and loaded (25 μL) onto a Dodecyl-sulfate-10% polyacrylamide gel and run during 1 h at 150 V. Proteins were transferred to PVDF membranes using the iBlot® standard P0 program. The membrane was blocked for 30 min in 5% dry-milk solution in PBS, washed and incubated with a 1:5000 dilution of a rabbit anti-TAP polyclonal antibody (Thermo Scientific®) for 1 h at room temperature, then with a 1:10,000 dilution of an anti-rabbit HRP-conjugate (Thermo Scientific®) for 1 h at room temperature. The membrane was then washed and developed with enhanced chemiluminescent detection reagents (ECL kit, GE Healthcare). After immunological detection the membrane was washed twice with $H_2O$ and subsequently stained with 0.1% (w/v) amido black in 4% (v/v) ethanol and 1% (v/v) glacial acetic acid.

**Phylogenetic analyses**. The alignment was done using MUSCLE v3.8.311[69] with default parameters. The software trimAl v32[70] was then used to trim the alignments generated by MUSCLE, with—automated1 option (implements a heuristic to decide the most appropriate mode depending on the alignment characteristics). We then used ProTest v2.43[71] to choose the best protein substitution models for each alignment, namely JTT + I + G + F. A maximum likelihood analysis was

conducted with PHYML v3.0.14[72] to reconstruct phylogenetic trees; support for the branches were determined from bootstrap analysis of 100 resampled datasets.

**Mating experiments**. Opaque cells were obtained from arg- and his- rme1 KO mutants (CEC5914 and CEC5917, respectively) as well as CEC5919 and CEC5921, arg- and his- derivatives of the reference strain SN76, respectively, by plating on L-sorbose[73]. Single colonies were transferred on SD plates prior to analytical PCR to check the mating-type locus status with primers pairs MTLa1-F and MTLa1-R or MTLalpha2-F and MTLalpha2-R. arg- *MTL**a*** and his- *MTL**α***-homozygous rme1- or SN76-derivatives were mixed in an equal proportion of $2 \times 10^7$ cells, spotted on Spider agar plates and incubated for 48 h at 25 °C and 5% $CO_2$. Mating mixtures were resuspended in water, diluted and spotted on selectable SD plates without arginine, without histidine or without both amino acids to quantify mating frequencies: rme1 *MTL**a***: SN76 *MTL**α*** = $(3.0 \pm 0.9) \times 10^{-1}$; SN76 *MTL**a***: SN76 *MTL**α*** = $(3.5 \pm 1.2) \times 10^{-1}$; rme1 *MTL**a***: rme1 *MTL**α*** = $(3.3 \pm 0.7) \times 10^{-1}$.

**Statistics and reproducibility**. All the microscopy images presented in the paper are representative of at least five images across two to four independent experiments. The western blot was performed once on two independent clones. Two-tailed, unpaired Student's *t*-tests were used throughout the study, unless stated otherwise in the figure legends. All bar graphs represent mean ± SD and are overlaid with dot plots. Details regarding sample size and significance are given in the legends.

**Reporting summary**. Further information on research design is available in the Nature Research Reporting Summary linked to this article.

## Data availability
Transcriptomic and ChIP-chip raw data generated during this study are available in Gene Expression Omnibus repository with the accession numbers GSE142370 and GSE142159, respectively. Combined and normalized data are also available in the Supplementary Data. *C. buenavistaensis RME1* sequence is available in GenBank under accession number MK070497. The source data underlying Figs. 3b and 5b and Supplementary Figs. 1a, 2a, b, 5, 7 and 8 are provided as a Source Data file.

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

## Acknowledgements

This research was supported by grants from the Agence Nationale de la Recherche (KANJI, ANR-08-MIE-033-01 to C.d.E. and F.D.; CANDIHUB, ANR-14-CE-0018 to C.d.E.; IBEID, ANR-10-LABX-62-IBEID to C.d.E.), the European Commission (FIN-SysB, PITN-GA-2008-214004 to C.d.E.) and the French Government's Investissement d'Avenir program (Institut de Recherche Technologique BIOASTER, ANR-10-AIRT-03 to C.d.E. and F.D.). S.Z. is an Institut Pasteur International Network Affiliate Program Fellow. S.Z., L.v.W. and A.H.C. were the recipients of post-doctoral fellowships from the European Commission (FINSysB, PITN-GA-2008-214004 to SZ), the Agence Nationale de la Recherche (KANJI, ANR-08-MIE-033-01 to S.Z.; ERA-Net Infect-ERA, FUN-COMPATH, ANR-14-IFEC-0004 to A.H.C.; CANDIHUB, ANR-14-CE-0018 to L.v.W.), the French Government's Investissement d'Avenir program (Institut de Recherche Technologique BIOASTER, ANR-10-AIRT-03 to S.Z. and A.H.C.) from the National Council of Science and Technology, Mexico to A.H.C. V.B. was supported by a grant from the Pasteur-Paris University (PPU) International PhD program and the "Fondation Daniel et Nina Carasso", ID belongs to the Pasteur-Paris University (PPU) International PhD program, which has received funding from the European Union's Horizon 2020 research and innovation programme under the Marie Sklodowska-Curie grant agreement No 665807, and from the Institut Carnot Pasteur Microbes & Santé.

## Author contributions

C.d.E., S.B.B., S.Z., L.v.W. and A.H.C. designed the study. A.H.C., V.B., L.v.W., I.D., S.B.B. performed molecular biology, microbiology and microscopy experiments. A.H.C. and S.Z. performed transcript profiling and ChIP-chip experiments as well as bioinformatic analyses. A.H.C., S.Z., V.B., L.v.W. performed qPCR. A.H.C. performed the screening and regulation experiment. N.S., M.E.B., F.B., L.B. and F.D. performed mouse experiments. G.M. and D.S. created mutant strains of *C. dubliniensis*. T.B., Y.Y. and Z.L. performed *C. buenavistaensis* sequencing. JR performed phylogenetic analyses. A.H.C., S.Z., S.B.B., L.v.W. and C.d.E. wrote the manuscript. All authors reviewed and approved the final version of the text.

## Competing interests

The authors declare no competing interests.
