## [Peer Review File · Nature Communications]

Reviewers' comments:

Reviewer #1 (Remarks to the Author):

This manuscript presents a very strong case that the transcription factor Rme1 of *Candida albicans* is a determinative activator or master regulator of chlamyospore formation. Genomic evidence includes ChIP-Chip and microarray data that reveals Rme1 to be a direct activator of chlamyospore associated genes. Functional evidence includes converse effects of a knockout allele (leading to no chlamyospore formation) and a hypermorphic high-level expression allele (leading to hyper-chlamyospore formation that bypasses genetic and environmental signals). Functional results are extended to two closely related *Candida* species that are also chlamyospore formers.

Experiments are generally very well designed. The use of microarrays rather than sequence-based analysis for ChIP and expression measurements is a little dated. However, major expression results are validated by additional approaches, and as for ChIP data this group is masterful based on many previous well confirmed papers. I especially appreciated the fact that fluorescent reporters were used to unambiguously identify chlamyospores, a task that can be challenging (as in the *efg1 RME1-up* strain). In my opinion there are no significant weaknesses in the data or logic.

One idea might be extended to broaden the interest in this manuscript. The authors argue very reasonably that Rme1 may have an antagonistic effect on hyphal morphogenesis (lines 469-470). More direct evidence for this inference, through phenotypic or gene expression data, would make the study a little bit more exciting.

The only important criticism I have of the manuscript is that the process of chlamyospore formation is of questionable biological significance at this time. The authors provide a balanced and accurate view of this issue in the Introduction, and present the findings in the broader context of fungal spore formation ability and *Candida albicans* cell type regulators in the Discussion. However, the data in this manuscript do not provide much insight into the evolutionary question of how Rme1 orthologs are repurposed as diverse sporulation regulators. Instead the conclusion provides a new evolutionary data point that adds to the question of how repurposing has occurred.

Minor points.

Lines 397-399 (Overexpression of SFL1...) This critically important sentence seems to be missing a few words and should be repaired. I personally found the images in Figure 6D to be less persuasive than most other micrographs in the manuscript. Therefore a clear statement from the authors of their conclusion seems to be quite important.

Lines 437-438. "Similarly, activation of IME1 expression commits *Saccharomyces cerevisiae* to enter meiosis and produce sexual spores." This statement conveys accurately what was deduced in the 1988 citation. However, many later papers indicate that IME1 expression commits cells to recombination and premeiotic S phase, but not to meiotic division and spore formation. Those later steps are controlled by a different regulatory mechanism. See the review "Sporulation in the Budding Yeast *Saccharomyces cerevisiae*" by Aaron M. Neiman, 2011, especially Figure 1.

Reviewer #2 (Remarks to the Author):

The formation of chlamyospores by *Candida albicans* on specific growth media is a well-known characteristic that is used for the identification of this pathogenic yeast (and the closely related species *C. dubliniensis*) in clinical microbiology laboratories. Several positive and negative

regulators of chlamyospore formation in *C. albicans* have been identified in the past decades, but the biological function of this cell type has remained elusive.

In the present study, the authors analysed the function of the transcription factor Rme1, which in *Saccharomyces cerevisiae* regulates meiosis, a process that has not been observed in *C. albicans*. They found that Rme1 binds to many genes that are upregulated during chlamyospore formation, implying a role in this developmental program. Forced overexpression of RME1 induced the expression of these genes and chlamyospore formation. Conversely, the expression of chlamyospore genes and chlamyospore formation under inducing conditions was abolished in an *rme1Δ* mutant. This was also the case in two other chlamyospore-forming species, *C. dubliniensis* and *C. buenavistaensis*. The RME1 gene from the latter two species, but not from *C. tropicalis* and *C. parapsilosis*, which do not produce chlamyospores, restored chlamyospore formation in a *C. albicans rme1Δ* mutant, indicating that species-specific differences in the Rme1 protein underlie its role in the regulation of this program. This study clearly establishes Rme1 as another important regulator of chlamyospore formation (besides other functions) in *C. albicans*.

Comments

1) The authors suggest that Rme1 is the key regulator of chlamyospore formation in *C. albicans*, also based on the observation that forced overexpression of RME1 bypasses the chlamyospore formation defect of *efg1Δ* and *hog1Δ* mutants. Interestingly, overexpression of the positive regulator SFL1 strongly increased chlamyospore formation (Fig. 6D) without increasing RME1 expression (Fig. S6). Similarly, deletion of the negative regulator NDT80 also increased chlamyospore formation (Fig. 6D) without increasing RME1 expression (Fig. S6). These results raise questions about the relative importance of the various positive and negative regulators in the control of chlamyospore development. More insight into the position of each transcription factor in the regulatory network controlling chlamyospore formation could be gained from the analysis of additional double mutants. Would the absence of the negative regulators Ndt80 and Nrg1 bypass the requirement for Rme1, i.e. would *rme1Δ ndt80Δ* and *rme1Δ nrg1Δ* double mutants still produce chlamyospores under inducing conditions? Furthermore, would RME1 overexpression in an *sfl1Δ* mutant still promote chlamyospore formation, as expected from the authors' model? Vice versa, would SFL1 overexpression allow chlamyospore formation in the absence of Rme1?

2) The correlation between RME1 expression levels and chlamyospore formation in different *C. albicans* strains is taken as evidence for a central role of Rme1 in the regulation of chlamyosporulation in *C. albicans* (that chlamyospore formation and expression of chlamyospore-specific genes such as CSP1 and CSP2 are correlated is not surprising). What about other regulators of chlamyospore formation? There might be a similar correlation, for example decreased NRG1 expression in strains that produce chlamyospores as compared with strains that grow as yeasts under the same conditions (see Fig. 3).

3) To identify genes that are differentially expressed upon doxycycline-induced RME1 expression, the authors compared gene expression in the presence and absence of doxycycline. This is formally incorrect, because some alterations in gene expression could be caused by doxycycline, and not by forced RME1 expression. Indeed, it has been reported that doxycycline affects gene expression in *C. albicans* and has phenotypic effects, especially at the relatively high concentration used in the experiments in the present study. The authors state that doxycycline did not significantly alter gene expression and refer to data in a Gene Expression Omnibus repository (lines 123-124). Although I trust the authors' assertion that most of the gene expression alterations observed in these experiments are caused by Rme1, I think that expression data reported in the paper should be based on the comparison of the RME1-overexpressing strain and a strain carrying a control construct (or at least the wild-type parental strain) in the presence of doxycycline. In other experiments in which RME1 expression was induced by doxycycline, control strains should also be grown in the presence of doxycycline.

4) Line 57: In my opinion, the term "epigenetic switch" is not appropriate to describe the

observation that chlamyospore formation, once induced, is completed even when the inducing signal is removed. Epigenetics describes phenotypic alterations that are inherited by progeny cells in the absence of changes in the genome sequence. When *C. albicans* chlamyospores germinate, they do not again produce chlamyospores in the absence of inducing signals, i.e., the morphology is not inherited by their progeny.

5) Line 67: haploid *S. cerevisiae* cells can mate and go through the sexual cycle more than once (as implied by the description "one more time").

6) Line 170: No information is provided on how mating efficiency and virulence of the *rme1Δ* mutant were tested.

7) Figure S3 is not very useful in its present form. Differences between the proteins should be highlighted somehow. Furthermore, amino acid positions should be indicated and the overall similarity between the proteins given.

8) Line 552: The authors should point out that the YPD medium used in their experiments contained only half the glucose concentration of standard YPD medium; otherwise, this information will be easily overlooked.

9) Lines 783-784: The scoring of chlamyospore formation (amount of chlamyospores found per field) appears somewhat arbitrary. Would it not be better to use the proportion of chlamyospores among all cells, considering that there might be differences in the total amount of cells per field when comparing different strains?

Reviewer #3 (Remarks to the Author):

This manuscript from Hernando-Cervantes et al. describes the identification of Rme1 as a key transcription factor regulating chlamyospore formation in *Candida*. The authors use CHIP-seq to identify an enrichment of Rme1 binding in the promoters of chlamyospore-induced genes. They demonstrate that overexpression of RME1 induces chlamyospores while deletion of the gene blocks chlamyospore formation. The ability to induce chlamyospores when overexpressed is only seen for RME1 genes from chlamyospore-forming *Candida* species. Moreover, the ability of different *C. albicans* isolates to form chlamyospores correlates with the level of RME1 expression in different strains. The authors also demonstrate that RME1 acts downstream of previously reported regulators of chlamyosporulation and outline the transcription factor network acting upstream of RME1 expression. Together, these observations convincingly place RME1 at the center of a regulatory circuit controlling this morphogenetic pathway.

Other comments:

A quick inspection of the previously reported (Pilage et al.) list of chlamyospore-induced and chlamyospore-repressed transcripts suggests that the overlap between these and the RME1 induced/repressed genes identified here is actually even better than the authors indicate. It would be useful to have some specific numbers (eg percent overlap) or statement of statistical significance.

I am somewhat mystified by the data in figure 2C. What is the significance of the Eosin Y and BODIPY staining? The authors indicate that these are hallmarks of chlamyospores but the references cited are for the use of these stains in other organisms. Unless there is evidence for these stains highlighting chlamyospores (BODIPY stains lipid droplets – don't all cells stain with

BODIPY?), this should be dropped. The appearance of Csp1-GFP and Ppga55-GFP specifically in the round cells at the end of the pseudohyphae is good evidence that these are chlamydozoospores and not simply aberrant cells produced by RME1 overexpression.

Answer to reviewer comments

Reviewer #1 (Remarks to the Author):

This manuscript presents a very strong case that the transcription factor Rme1 of *Candida albicans* is a determinative activator or master regulator of chlamyospore formation. Genomic evidence includes ChIP-Chip and microarray data that reveals Rme1 to be a direct activator of chlamyospore-associated genes. Functional evidence includes converse effects of a knockout allele (leading to no chlamyospore formation) and a hypermorphic high-level expression allele (leading to hyper-chlamyospore formation that bypasses genetic and environmental signals). Functional results are extended to two closely related *Candida* species that are also chlamyospore formers.

Experiments are generally very well designed. The use of microarrays rather than sequence-based analysis for ChIP and expression measurements is a little dated. However, major expression results are validated by additional approaches, and as for ChIP data this group is masterful based on many previous well confirmed papers. I especially appreciated the fact that fluorescent reporters were used to unambiguously identify chlamyospores, a task that can be challenging (as in the *efg1 RME1-up* strain). In my opinion there are no significant weaknesses in the data or logic.

One idea might be extended to broaden the interest in this manuscript. The authors argue very reasonably that Rme1 may have an antagonistic effect on hyphal morphogenesis (lines 469-470). More direct evidence for this inference, through phenotypic or gene expression data, would make the study a little bit more exciting.

We thank Reviewer#1 for the positive comments. We totally agree with Reviewer#1 that a deeper investigation of Rme1 function in hyphal morphogenesis would be an exciting future avenue of research. Based on our genome-wide location and expression analyses, it is clear that Rme1 directly controls the expression of many genes involved in filamentous growth (Figure 1c). Because chlamyospore formation and filamentous growth are interconnected processes, we believe that a fine-tuned analysis of Rme1 genome occupancy together with the associated transcriptional changes under various filamentation-stimulating conditions would provide a sharper idea of Rme1 function in hyphal development. By combining these data with extensive phenotypic experiments in various genetic backgrounds, we would be able to discriminate – to some extent – the responses specifically linked to chlamyospore formation from the ones directly linked to filamentous growth and determine whether indeed Rme1 antagonistically affects hyphal morphogenesis. We are eager to answer to this question in a separate future study. We thank again this reviewer for raising this excellent point.

The only important criticism I have of the manuscript is that the process of chlamyospore formation is of questionable biological significance at this time. The authors provide a balanced and accurate view of this issue in the Introduction, and present the findings in the broader context of fungal spore formation ability and *Candida albicans* cell type regulators in the Discussion. However, the data in this manuscript do not provide much insight into the evolutionary question of how Rme1 orthologs are repurposed as diverse sporulation regulators. Instead the conclusion provides a new evolutionary data point that adds to the question of how repurposing has occurred.

We thank Reviewer #1 for this comment. Accordingly, we have added the following paragraph to discuss this aspect further (Lines 508-532): “Our comparative analyses of Rme1 homologs from the *Candida* clade as well as those from additional representative species of the ascomycete lineage show low-to-moderate sequence divergence within the DNA-binding domain (DBD, located at the C-terminus); but high sequence divergence in the N-terminal 2/3rds of the proteins from *C. albicans*, *C. dubliniensis*, *C. buenavistaensis* and *C. tropicalis*. Intriguingly, the Rme1 N-termini in *C. albicans*, *C. dubliniensis* and *C. buenavistaensis* carry extensive repeats of poly-Asn/Gln/His stretches that are absent in other species from the *Candida* clade. More precisely, while *C. albicans* Rme1 DBD displays high sequence similarity with the equivalent sequences from the phylogenetically-related species *C. dubliniensis*, *C. buenavistaensis*, *C. tropicalis*, *L. elongisporus*, *C. orthopsilosis*, *C. parapsilosis* and *C. metapsilosis*, sequence similarity drastically decreases in the N-terminal 2/3rds of the proteins when we compare Rme1 sequences from *C. albicans*, *C. dubliniensis*, *C. buenavistaensis* and *C. tropicalis* to those from *L. elongisporus*, *C. orthopsilosis*, *C. parapsilosis* and *C. metapsilosis*. This suggests that species-specific differences in the Rme1 protein underlie its role in the regulation of chlamyospore formation and that functional repurposing might have

occurred - at least in part - through substitutions affecting the N-terminal half of the protein, which may include interaction domains with additional (co-)regulators/regulatory proteins and/or the RNA-polymerase II holoenzyme. We have also performed motif discovery analyses to determine if *C. albicans* Rme1 binds to a divergent *cis*-acting element. We did not identify regular patterns that could be indicative of motif conservation and/or overrepresentation in *C. albicans* Rme1 target promoters. Alternatively, a search of the motif identified in *S. cerevisiae* (5'-GWACCWCAADA-3') did not return any significant matches/positive hits. We think that functional repurposing might have occurred through alteration of both Rme1 functional domains and *cis*-acting elements. Our future work will be complemented by extensive bioinformatic analyses to delineate the mechanistic aspects that potentially led to neofunctionalization of *C. albicans* Rme1."

Minor points.

Lines 397-399 (Overexpression of *SFL1*...). This critically important sentence seems to be missing a few words and should be repaired. I personally found the images in Figure 6D to be less persuasive than most other micrographs in the manuscript. Therefore a clear statement from the authors of their conclusion seems to be quite important.

We agree. We rephrased the sentence as follows: "In chlamyospore-inducing conditions, overexpression of *SFL1* led to chlamyosporulation, whereas *NDT80* overexpression abolished chlamyospore formation (Fig. 6d), confirming the antagonistic functions of these two transcription factors" (Lines 410-412). We also modified panels c and d in Figure 6 for more clarity.

Lines 437-438. "Similarly, activation of *IME1* expression commits *Saccharomyces cerevisiae* to enter meiosis and produce sexual spores." This statement conveys accurately what was deduced in the 1988 citation. However, many later papers indicate that *IME1* expression commits cells to recombination and premeiotic S phase, but not to meiotic division and spore formation. Those later steps are controlled by a different regulatory mechanism. See the review "Sporulation in the Budding Yeast *Saccharomyces cerevisiae*" by Aaron M. Neiman, 2011, especially Figure 1.

We thank Reviewer #1 for pointing this out. We rephrased the sentence and replaced the reference as follows: "Similarly, activation of *IME1* expression is the crux triggering *Saccharomyces cerevisiae* entry into the sporulation pathway⁴⁴" (Lines 481-482).

Reviewer #2 (Remarks to the Author):

The formation of chlamyospores by *Candida albicans* on specific growth media is a well-known characteristic that is used for the identification of this pathogenic yeast (and the closely related species *C. dubliniensis*) in clinical microbiology laboratories. Several positive and negative regulators of chlamyospore formation in *C. albicans* have been identified in the past decades, but the biological function of this cell type has remained elusive.

In the present study, the authors analysed the function of the transcription factor Rme1, which in *Saccharomyces cerevisiae* regulates meiosis, a process that has not been observed in *C. albicans*. They found that Rme1 binds to many genes that are upregulated during chlamyospore formation, implying a role in this developmental program. Forced overexpression of *RME1* induced the expression of these genes and chlamyospore formation. Conversely, the expression of chlamyospore genes and chlamyospore formation under inducing conditions was abolished in an *rme1Δ* mutant. This was also the case in two other chlamyospore-forming species, *C. dubliniensis* and *C. buenavistaensis*. The *RME1* gene from the latter two species, but not from *C. tropicalis* and *C. parapsilosis*, which do not produce chlamyospores, restored chlamyospore formation in a *C. albicans rme1Δ* mutant, indicating that species-specific differences in the Rme1 protein underlie its role in the regulation of this program. This study clearly establishes Rme1 as another important regulator of chlamyospore formation (besides other functions) in *C. albicans*.

Comments

1) The authors suggest that Rme1 is the key regulator of chlamyospore formation in *C. albicans*, also based on the observation that forced overexpression of *RME1* bypasses the chlamyospore formation defect of *efg1Δ* and *hog1Δ* mutants. Interestingly, overexpression of the positive regulator *SFL1* strongly increased

chlamydospore formation (Fig. 6D) without increasing *RME1* expression (Fig. S6). Similarly, deletion of the negative regulator *NDT80* also increased chlamydospore formation (Fig. 6D) without increasing *RME1* expression (Fig. S6). These results raise questions about the relative importance of the various positive and negative regulators in the control of chlamydospore development. More insight into the position of each transcription factor in the regulatory network controlling chlamydospore formation could be gained from the analysis of additional double mutants. Would the absence of the negative regulators Ndt80 and Nrg1 bypass the requirement for Rme1, *i.e.* would *rme1Δ ndt80Δ* and *rme1Δ nrg1Δ* double mutants still produce chlamydospores under inducing conditions? Furthermore, would *RME1* overexpression in an *sf11Δ* mutant still promote chlamydospore formation, as expected from the authors' model? Vice versa, would *SFL1* overexpression allow chlamydospore formation in the absence of Rme1?

We thank Reviewer #2 for suggesting these important additional experiments. We have created the *ndt80ΔΔ rme1ΔΔ* double mutant and have also overexpressed *RME1* and *SFL1* in *sf11ΔΔ* and *rme1ΔΔ* genetic backgrounds, respectively. The resulting data are presented in new Supplementary figure 6. We found that a functional *RME1* gene is required for *SFL1* to induce chlamydospore formation upon overexpression (New Supplementary Figure 6a, upper panel), whereas overexpression of *RME1* does not require a functional *SFL1* to activate the chlamydospore program (New Supplementary figure 6a, lower panel), indicating that *RME1* acts downstream of *SFL1*. On the other hand, deletion of *RME1* in the *ndt80ΔΔ* genetic background abolished chlamydospore development (New Supplementary figure 6b), indicating that *NDT80* loss-of-function does not bypass the requirement for *RME1* to produce chlamydospores; further highlighting the central role of *RME1* in this developmental program. We described these new data in lines 413-422.

2) The correlation between *RME1* expression levels and chlamydospore formation in different *C. albicans* strains is taken as evidence for a central role of Rme1 in the regulation of chlamydospore formation in *C. albicans* (that chlamydospore formation and expression of chlamydospore-specific genes such as *CSP1* and *CSP2* are correlated is not surprising). What about other regulators of chlamydospore formation? There might be a similar correlation, for example decreased *NRG1* expression in strains that produce chlamydospores as compared with strains that grow as yeasts under the same conditions (see Fig. 3).

We thank Reviewer #2 for suggesting these additional experiments. We measured the expression levels of both *NRG1* (negative regulator of chlamydospore formation) and *SFL1* (positive regulator of chlamydospore formation) in clinical isolates displaying weak (CEC1424, CEC1426) and strong (CEC3620, CEC2018) chlamydospore formation (Supplementary Figure 8). In agreement with Reviewer #2 and our expectations, the expression levels of *SFL1* and *NRG1* correlated and inversely correlated, respectively, with the ability of clinical isolates to form chlamydospores. Consistent with the role of *NRG1* and *SFL1* as a negative and positive regulators of chlamydospore formation, their expression levels were unchanged (*SFL1*) and increased (*NRG1*) in weak chlamydospore formers and were increased (*SFL1*) and decreased (*NRG1*) in strong chlamydospore formers. We have included these important additional findings in a new section of the manuscript entitled "Regulators of chlamydospore development other than Rme1 display gene expression patterns correlating with the efficiency of *C. albicans* natural isolates to form chlamydospores" (Lines 428-445).

3) To identify genes that are differentially expressed upon doxycycline-induced *RME1* expression, the authors compared gene expression in the presence and absence of doxycycline. This is formally incorrect, because some alterations in gene expression could be caused by doxycycline, and not by forced *RME1* expression. Indeed, it has been reported that doxycycline affects gene expression in *C. albicans* and has phenotypic effects, especially at the relatively high concentration used in the experiments in the present study. The authors state that doxycycline did not significantly alter gene expression and refer to data in a Gene Expression Omnibus repository (lines 123-124). Although I trust the authors' assertion that most of the gene expression alterations observed in these experiments are caused by Rme1, I think that expression data reported in the paper should be based on the comparison of the *RME1*-overexpressing strain and a strain carrying a control construct (or at least the wild-type parental strain) in the presence of doxycycline. In other experiments in which *RME1* expression was induced by doxycycline, control strains should also be grown in the presence of doxycycline.

We agree with these comments from Reviewer #2. We performed genome-wide transcript profiling of a strain carrying an empty vector (*i.e.* *P_{TET}* only) and compared the transcriptome of the strain grown for 4 hours in the absence versus in the presence of 40 μg/ml of doxycycline (See Figure S3 and Table S7 in Znaidi *et al.* 2018 Cell Microbiol.20(11):e12890). As stated in the published work, only 13 and 9 genes were upregulated

and downregulated, respectively, with maximum fold change values of 2.0 and -2.5, respectively (Table S7, Znaidi *et al.* 2018 Cell Microbiol.20(11):e12890), including 10 genes displaying increased fold change due to a skew in the distribution of high intensity signal (Figure S3B and Table S7, Znaidi *et al.* 2018 Cell Microbiol.20(11):e12890). Furthermore, we have previously shown that adding doxycycline or an alternative tetracycline derivative (anhydrotetracycline, Basso *et al.* 2017 Mol. Microbiol. 106(1):157-182) to the medium does not significantly alter gene expression. We agree that comparing the phenotypes to a control strain in the presence of doxycycline would be a nice additional control, but we think that using strains with the exact same background is of importance when working with *C. albicans*, to minimize strain-specific effects.

4) Line 57: In my opinion, the term “epigenetic switch” is not appropriate to describe the observation that chlamyospore formation, once induced, is completed even when the inducing signal is removed. Epigenetics describes phenotypic alterations that are inherited by progeny cells in the absence of changes in the genome sequence. When *C. albicans* chlamyospores germinate, they do not again produce chlamyospores in the absence of inducing signals, *i.e.*, the morphology is not inherited by their progeny. We thank Reviewer #2 for providing this important comment. We modified the sentence in lines 57-59 as follows: “We suggest that Rme1 function spans from the regulation of meiosis in sexual yeasts to the control of a developmental program necessary for asexual spore formation in meiosis-defective *Candida* species”.

5) Line 67: haploid *S. cerevisiae* cells can mate and go through the sexual cycle more than once (as implied by the description “one more time”). We modified the sentence: “(...) upon improvement of the environmental conditions will develop into Mata and Mata haploid cells, able to mate and go through the sexual cycle” (Lines 67-68).

6) Line 170: No information is provided on how mating efficiency and virulence of the *rme1Δ* mutant were tested. We added the description of both experiments in the METHODS section (Lines 984-994 and 716-726).

7) Figure S3 is not very useful in its present form. Differences between the proteins should be highlighted somehow. Furthermore, amino acid positions should be indicated and the overall similarity between the proteins given. We agree. We modified the figure accordingly (amino acid numbering as well as indication of identical and similar residues).

8) Line 552: The authors should point out that the YPD medium used in their experiments contained only half the glucose concentration of standard YPD medium; otherwise, this information will be easily overlooked. We thank Reviewer #2 for pointing out this error. We used standard YPD throughout the study, and corrected the mistake.

9) Lines 783-784: The scoring of chlamyospore formation (amount of chlamyospores found per field) appears somewhat arbitrary. Would it not be better to use the proportion of chlamyospores among all cells, considering that there might be differences in the total amount of cells per field when comparing different strains? Reviewer #2 raises an important point here. We used the scoring method by Böttcher *et al.* 2016 Front Microbiol. 7:1697, as there is no way to actually measure accurately the proportion of chlamyospores in the cell population.

Reviewer #3 (Remarks to the Author):

This manuscript from Hernando-Cervantes *et al.* describes the identification of Rme1 as a key transcription factor regulating chlamyospore formation in *Candida*. The authors use CHIP-seq to identify an enrichment of Rme1 binding in the promoters of chlamyospore-induced genes. They demonstrate that overexpression of *RME1* induces chlamyospores while deletion of the gene blocks chlamyospore formation. The ability to induce chlamyospores when overexpressed is only seen for *RME1* genes from chlamyospore-forming *Candida* species. Moreover, the ability of different *C. albicans* isolates to form chlamyospores correlates with the level of *RME1* expression in different strains. The authors also demonstrate that *RME1* acts

downstream of previously reported regulators of chlamydosporulation and outline the transcription factor network acting upstream of *RME1* expression. Together, these observations convincingly place *RME1* at the center of a regulatory circuit controlling this morphogenetic pathway.

Other comments:

A quick inspection of the previously reported (Pilage *et al.*) list of chlamyospore-induced and chlamyospore-repressed transcripts suggests that the overlap between these and the *RME1* induced/repressed genes identified here is actually even better than the authors indicate. It would be useful to have some specific numbers (eg percent overlap) or statement of statistical significance.

We thank Reviewer #3 for this comment. We looked at the occurrence of genes that are both bound and upregulated by Rme1 (257 out of 6,083 *C. albicans* genes represented on the arrays, 4.2%, Figure 1) among the set of genes that are upregulated ≥ 2 -fold in the *C. albicans nrg1Δ* mutant compared to the wild-type strain grown in the chlamyospore-inducing Staib medium (Palige *et al.*, 2013). We found 34 occurrences out of 271 genes upregulated in the *nrg1Δ* strain vs. wild-type (12.5%), yielding a ~ 3 -fold enrichment of Rme1 direct targets in Palige *et al.* datasets ($P = 6.72 \times 10^{-9}$ using a hypergeometric test). We added this statement in lines 161-167.

I am somewhat mystified by the data in figure 2C. What is the significance of the Eosin Y and BODIPY staining? The authors indicate that these are hallmarks of chlamydo-spores but the references cited are for the use of these stains in other organisms. Unless there is evidence for these stains highlighting chlamydo-spores (BODIPY stains lipid droplets – don't all cells stain with BODIPY?), this should be dropped. The appearance of Csp1-GFP and Ppga55-GFP specifically in the round cells at the end of the pseudohyphae is good evidence that these are chlamydo-spores and not simply aberrant cells produced by *RME1* overexpression.

We agree with Reviewer #3 that the appearance of Csp1-GFP and Ppga55-GFP specifically in the round cells at the end of the pseudohyphae is good evidence that these are chlamydo-spores. However, because of the high content of lipid droplets in chlamydo-spores (the measured free lipid content represents 12% of chlamydo-spore dry weight), we used BODIPY as an additional stain to further highlight the nature of the observed cellular components in the cytoplasm of chlamydo-spores (*i.e.* lipid droplets) and appraise additional anatomical features of this developmental program, such as the chlamydo-spore cell wall which is particularly enriched in chitosan (revealed by staining with Eosin Y). The difference in the content of lipids is also remarkably noticeable between the SC5314 strain and the *RME1*-overexpressing strain (Figure 2c, upper and middle left panels). We modified the statement describing BODIPY and Eosin Y staining for more clarity as follows (lines 190-193): "EosinY, which detects chitosan in the chlamydo-spore cell wall and the lipid droplet-specific fluorescent dye BODIPY, also stained the rounded cells produced by the WT and *P_{TET}-RME1* strains, further highlighting features associated with chlamydo-spore formation (Fig. 2c). We hope that Reviewer #3 still appreciates the different staining approaches that we used in our work.

REVIEWERS' COMMENTS

Reviewer #2 (Remarks to the Author):

In their revised manuscript, the authors have adequately addressed my previous comments. I have some minor additional comments.

1) The authors should state in which strain background the new epistasis experiments were performed, both in the text (page 21 and methods) and in the legend to Figure S6 (as for the other experiments). From Table S1 it seems that the original parent of the new strains was an auxotrophic derivative of strain SC5314 (SN76), but finding this information required going through all the previous parental strains in the supplementary table, and this is inconvenient for readers. It is relevant to know that these tests were performed with a poor chlamyospore producer.

2) The methods section also describes an *nrg1 rme1* double mutant (lines 898-899, also listed in Table S1), but results with this strain are not included in Figure S6b and not described in the text. It would certainly be interesting to know the phenotype of this double mutant.

Reviewer #3 (Remarks to the Author):

It's a somewhat minor point for an otherwise very nice study, but the authors have not directly addressed my concern about their staining. Certainly if Eosin Y and BODIPY are established ways of distinguishing chlamyospores it makes sense to include the data. But references for these observations should be cited - the two references given (one for chitosan and one for BODIPY) describe the staining techniques, but not that these can be used to identify chlamyospores.

In their response, the authors note that "the measured free lipid content represents 12% of chlamyospore dry weight". If so, that certainly would be consistent with the BODIPY staining. The reference for this fact, and the logic therefore of staining for lipid droplets should be included in the text. If similar evidence for Eosin Y staining has been published, it should be cited as well.

REVIEWERS' COMMENTS

Reviewer #2 (Remarks to the Author):

In their revised manuscript, the authors have adequately addressed my previous comments. I have some minor additional comments.

1) The authors should state in which strain background the new epistasis experiments were performed, both in the text (page 21 and methods) and in the legend to Figure S6 (as for the other experiments). From Table S1 it seems that the original parent of the new strains was an auxotrophic derivative of strain SC5314 (SN76), but finding this information required going through all the previous parental strains in the supplementary table, and this is inconvenient for readers. It is relevant to know that these tests were performed with a poor chlamyospore producer.

Indeed the mutants strains were all derivatives of SN76. We have modified the text in the results section as follows: “

To further explore the relative importance of *SFL1*, *NDT80* and *RME1* in chlamyospore formation, we tested the epistatic relationship between the three regulators in a SC5314 strain-derivative (Supplementary Fig. 6)” (lines 320-333).

We also added the information in methods section: « The transient CRISPR-Cas9 system was also used to construct double mutants of *NDT80* or *NRG1* in the *rme1* mutant strain CEC5914, deriving from SN76. » (lines 755-756).

2) The methods section also describes an *nrg1 rme1* double mutant (lines 898-899, also listed in Table S1), but results with this strain are not included in Figure S6b and not described in the text. It would certainly be interesting to know the phenotype of this double mutant.

We modified the last paragraph of the results section to include this result : « We confirmed the phenotype of the *nrg1* mutant (Supplementary Fig. 6c), and showed that deleting *RME1* in the *nrg1* background abolished chlamyospore formation (Supplementary Fig. 6c) » (lines 344-346). We also added a panel to Supplementary Fig. 6.

Reviewer #3 (Remarks to the Author):

It's a somewhat minor point for an otherwise very nice study, but the authors have not directly addressed my concern about their staining. Certainly if Eosin Y and BODIPY are established ways of distinguishing chlamyospores it makes sense to include the data. But references for these observations should be cited - the two references given (one for chitosan and one for BODIPY) describe the staining techniques, but not that these can be used to identify chlamyospores.

In their response, the authors note that "the measured free lipid content represents 12% of chlamyospore dry weight". If so, that certainly would be consistent with the BODIPY staining. The reference for this fact, and the logic therefore of staining for lipid droplets should be included in the text. If similar evidence for Eosin Y staining has been published, it should be cited as well.

We have modified the end of the paragraph to explain our rationale for using the dyes, and added the requested reference. « We then reasoned that the high lipid content of chlamyospores²⁷ would allow staining by BODIPY, a lipophilic fluorescent dye²⁸. Similarly,

EosinY, which detects chitosan, a component of *S. cerevisiae*'s ascospores- and *C. neoformans* chlamydospores-cell wall²⁹, and does not stain viable *C. albicans* blastospores³⁰, could detect specific features of the *C. albicans* chlamydospores cell wall. Both EosinY and BODIPY stained the rounded cells produced by the WT and P_{TET-RME1} strains, further highlighting features associated with chlamydospore formation (Fig. 2c), thus allowing confirmation of chlamydospore formation in clinical isolates. » (lines 179-186).